# Sympathetic nervous system controls resolution of inflammation via regulation of repulsive guidance molecule A

Andreas Körner[1], Martin Schlegel[1], Torsten Kaussen[2], Verena Gudernatsch[1], Georg Hansmann[2], Timo Schumacher[2], Martin Giera [3] & Valbona Mirakaj[1]

The bidirectional communication between the immune and nervous system is important in regulating immune responses. Here we show that the adrenergic nerves of sympathetic nervous system orchestrate inflammation resolution and regenerative programs by modulating repulsive guidance molecule A (RGM-A). In murine peritonitis, adrenergic nerves and RGM-A show bidirectional activation by stimulating the mutual expression and exhibit a higher potency for the cessation of neutrophil infiltration; this reduction is accompanied by increased pro-resolving monocyte or macrophage recruitment, polymorphonucleocyte clearance and specialized pro-resolving lipid mediators production at sites of injury. Chemical sympathectomy results in hyperinflammation and ineffective resolution in mice, while RGM-A treatments reverse these phenotypes. Signalling network analyses imply that RGM-A and β2AR agonist regulate monocyte activation by suppressing NF-κB activity but activating RICTOR and PI3K/AKT signalling. Our results thus illustrate the function of sympathetic nervous system and RGM-A in regulating resolution and tissue repair in a murine acute peritonitis model.

---

[1] Department of Anesthesiology and Intensive Care Medicine, Molecular Intensive Care Medicine, University Hospital Tübingen, Eberhard-Karls University, Hoppe-Seyler-Str. 3, 72076 Tübingen, Germany. [2] Department of Pediatric Cardiology and Critical Care, Carl-Neuberg-Str. 1, Hannover Medical School, 30625 Hannover, Germany. [3] Center for Proteomics and Metabolomics, Leiden University Medical Center (LUMC), Albinusdreef 2, 2333 ZA Leiden, The Netherlands. Correspondence and requests for materials should be addressed to V.M. (email: valbona.mirakaj@uni-tuebingen.de)

Acute inflammation is a fundamental process that underlies multiple physiological and pathological mechanisms. A critical step in the initial immune response is the control of leukocyte migration, and if it fails, chronic inflammation can occur, leading to collateral tissue destruction and the loss of functional organ integrity. Resolution of an acute inflammatory response is a fundamental phase during which specialized lipid mediators (SPMs) with pro-resolving functions, including lipoxins, resolvins, protectins, and maresins, are biosynthesized to resolve the tissue insult, clear the infiltrated inflammatory cells and ultimately restore tissue homeostasis[1]. At the cellular level, this resolution process relies on complex events, including the cessation of neutrophil influx, the counter regulation of pro-inflammatory mediators, apoptosis of polymorphonuclear cells (PMNs), and the active clearance of apoptotic cells and invading microorganisms. Cells such as macrophages (MΦ) are central regulators in the maintenance of tissue homeostasis and repair by switching their phenotype from pro- to anti-inflammatory/pro-healing.

A pattern for guidance cues exists in the developing nervous system where axons are accurately guided to their final location through a balance of chemoattractive or chemorepulsive signals. One such guidance protein, repulsive guidance molecule-A (RGM-A), a glycosylphosphatidylinositol (GPI)-linked membrane glycoprotein, mediates chemorepulsive signals to steer axonal growth cones to their targets in the brain[2,3]. Studies have shown notable guidance roles for RGM-A and its receptor neogenin during embryonic development and morphogenetic processes including cell adhesion, cell migration, cell polarity and cell differentiation[4,5]. Recent evidence identified RGM-A in peripheral tissues, where it was shown to play crucial roles in the onset of an acute inflammatory response and in the pathology of autoimmune encephalomyelitis[6–8]. In this context, an efficient immune response against invading pathogens and complete resolution of tissue inflammation are the ideal outcomes for the affected tissues to restore their functional integrity. Non-resolving inflammation can result in severe critical illness, as observed in pathologies such as peritonitis, respiratory distress syndrome or sepsis. Recent insights have revealed the bidirectional communication between the immune system and the nervous system to be important in regulating immunological mechanisms[9]. Particularly, the neuronal reflexes, sense peripheral inflammation, and arrange inflammatory events within the initiation of inflammation. Lately, we identified cholinergic nerve signaling to control the generation of immunoresolvents such as the neuronal guidance protein Netrin-1 and the SPMs during acute inflammation[10]. In light of these accumulated findings, we decided to address the role of sympathetic nervous system (SNS) combined with the immunomodulatory actions of RGM-A in regulating resolution mechanism.

In the present report, we find a dynamic adrenergic nerve—RGM-A cooperation in controlling inflammation-resolution programs. This reflects in the shift of the phenotype from classical (M1) to alternative (M2) phenotype in functional studies. Studies in a murine peritonitis model further show that both adrenergic nerves and RGM-A synergistically reduce the level of inflammatory peritonitis, shorten the resolution interval, stimulate the local generation of pro-resolving lipid mediators, promote the clearance of apoptotic cells and stimulate tissue regeneration. Chemical sympathectomy increases the severity of murine peritonitis and lowers resolution. Administration of RGM-A to chemically sympathectomized mice recovers the resolution tone. Protein microarray analysis reflects suppression of NF-κB, activation of RICTOR signaling and PI3K/AKT signaling in peritoneal monocytes following the stimulation with RGM-A and/or β2AR agonist. Together, these results show a new aspect of the neural-reflex circuit involving adrenergic nerves and RGM-A that controls key innate protective mechanisms in the resolution of acute inflammation and promotes tissue repair and regeneration.

## Results

**RGM-A controls the macrophage inflammatory phenotype.** Recent evidence indicates that the monocyte and macrophage lineage is of pivotal importance in tissue homeostasis and the resolution of inflammation[11–13]. We first analyzed RGM-A expression in human monocyte-derived MΦ that were differentiated to classically (M1) or alternatively (M2) by stimulation with GM-CSF or M-CSF, respectively, for 7 days and found higher RGM-A transcript in M2 MΦ than in M1 MΦ (Fig. 1a). The macrophage phenotype is a result of differentiation and polarization, depending on the exposed signal[12]. Since cell shapes mark the differentiation to the M1 or M2 phenotypes[14], we stimulated human peripheral blood mononuclear cells (PBMC) with GM-CSF, M-CSF analyzed the cell morphology (Supplementary Fig. 1a). Treatment with GM-CSF induced the M1 phenotype and a specific round shape, whereas M-CSF-activated M2 MΦ showed an elongated morphology (Fig. 1b). Surprisingly, activation with RGM-A induced the M2 phenotype with a significantly greater number of elongated cell shapes compared to round M1 cells (Fig. 1b). Next we profiled the expression of key genes contributing to M1/M2 differentiation. RGM-A decreased the levels of M1 markers, such as STAT-1 and CD80, and significantly increased the levels of the M2 markers Arg1 and CD163 (Fig. 1c), which are phagocytic receptors and markers of the anti-inflammatory and efferocytic M2 phenotype[15]. To investigate whether RGM-A may play a direct role in the phenotypic polarization of MΦ, we challenged M1 MΦ (GM-CSF cultured) with RGM-A and subsequently stimulated them with TNF-α or vehicle for 24 h. We observed a significant reduction in the levels of the M1 markers and pro-inflammatory cytokines whereas the M2 markers and the anti-inflammatory cytokine were significantly increased (Fig. 1d), Together, these data indicate that RGM-A induced differentiation and polarization toward the M2 pro-healing and pro-resolving phenotype[13].

**RGM-A alters PMN and MΦ chemotaxis and functions.** Because timing and dynamics of leukocyte responses are thought to be crucial for the progression and resolution of inflammation. we sought to investigate the effect of RGM-A on the regulation of neutrophil and MΦ chemotaxis and chemokinesis. We designed a special microfluidic chamber that allows real-time measurements of leukocyte chemotaxis in response to chemotactic substances (Fig. 1e and Supplementary Fig. 2). Chemoattractant gradients, such as N-formylmethionyl-leucyl-pheylalanine (fMLP), monocyte chemotactic protein (MCP-1) and RGM-A (±MCP-1), were established between a range of eight peripheral wells and a central cell loading well. As expected, neutrophil migration toward fMLP was strong, reaching a maximum after 4 h. RGM-A reduced the chemotaxis of PMNs to fMLP (Fig. 1e), but in the absence of the chemoattracting substance, RGM-A showed a slight independent effect on PMN migration, implying a repulsive impact on chemotaxis (Fig. 1e). However, M1 MΦ recruitment toward MCP-1 was slower than PMN recruitment, achieving a maximum at 8–9 h. A strong reduction in M1 MΦ chemotaxis in the direction of the MCP/RGM-A gradient was observed, where the cells migrated directly toward the peripheral chamber (Fig. 1f). Furthermore, we found that RGM-A displayed a significant influence on chemokinesis. The treatment with RGM-A indicated a significant increase in macrophage migration toward the RGM-A gradient, whereas the macrophage migration toward the MCP-1 gradient was not affected suggesting that RGM-A directly

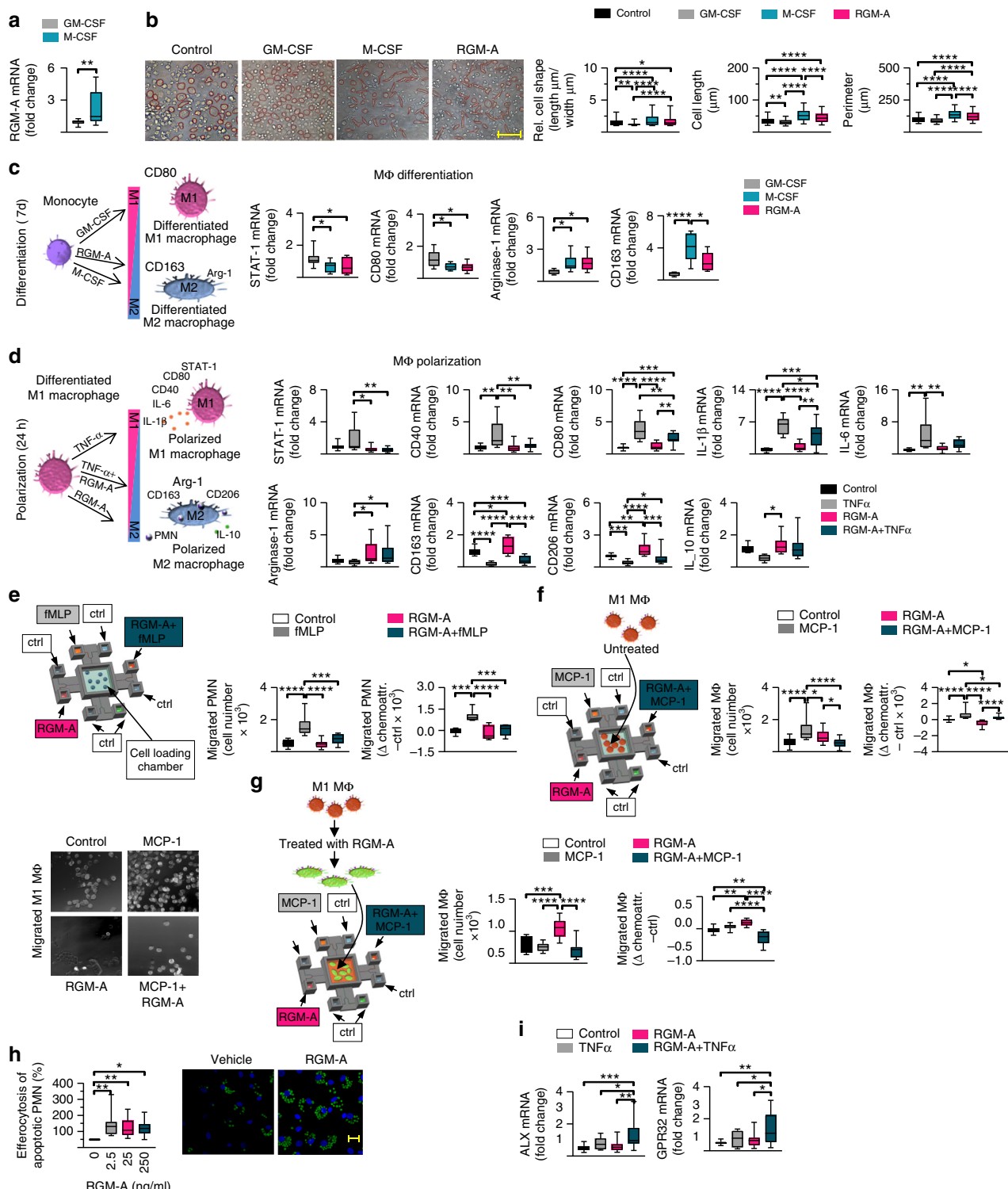

shifted the polarization state toward the M2 phenotype (Fig. 1g). We investigated the influence of RGM-A in clearance and found an increase in the capacity of primary human MΦ to uptake apoptotic human PMNs and serum-treated zymosan A (ZyA) (Fig. 1h). The G protein-coupled receptors ALX/FPR2 and GPR32 have been shown to mediate pro-resolving actions[1].We found RGM-A significantly to enhance GPR32 and ALX/FPR2 mRNA levels in human MΦ (Fig. 1i). These results indicate that RGM-A restored the chemotactic and chemokinetic responses of PMNs and MΦ and enhanced macrophage phagocytosis in vitro, and these effects can be defined as pro-resolving mediator responses[1].

**RGM-A$^{+/-}$ mice display impaired resolution of inflammation.** Based on the data described above, we determined the role of endogenous RGM-A by employing a well-established model of self-limited inflammation and monitoring the initiation and resolution phases[16–18]. Because RGM-A$^{-/-}$ mice are not viable, we injected heterozygous RGM-A-deficient (RGM-A$^{+/-}$) mice and their littermate controls with ZyA and collected peritoneal lavages at 4, 12, 24 and 48 h. In wild-type (WT) littermates, the acute infiltration of leukocytes was initiated during the initial phase of inflammation, with a maximal PMN infiltration at 4 h, followed by a decrease, giving a resolution interval (Ri) of ≈ 12 h (Fig. 2a).

**Fig. 1** RGM-A controls macrophage phenotype and regulates human PMN/macrophage chemotaxis and chemokinesis. To mark the differentiation into the classically activated (M1) or alternatively activated (M2) phenotype, human PBMCs were stimulated with GM-CSF, M-CSF or RGM-A for 7 d, and **a** RGM-A transcript expression in differentiated M1 and M2 macrophages (MΦ) was quantified by RT-PCR ($n = 12$). **b** The cell morphology was analyzed by phase contrast images and measurements of the cell shape, cell length and perimeter in each high-power field (magnification ×200), Scale bar: 20 μm ($n = 220$). **c** The expression of key genes that contribute to the M1 differentiation, *STAT-1* and *CD80*, and central genes of the M2 differentiation, *Arg1* and *CD163*, were analyzed ($n = 12$). **d** To investigate the role of RGM-A in the phenotypic polarization of macrophages, M1 macrophages were challenged with RGM-A and subsequently with TNF-α or vehicle for 24 h. The gene expression of M1 polarization markers including *STAT-1, CD40, CD80, IL-1β*, and *IL-6* as well as key genes of the M2 polarization such as *Arg1, CD163, CD206*, and *IL-10* were quantified by RT-PCR ($n = 14$). **e** Schematic model of the microfluidic migration chamber. To investigate the effect of RGM-A on the regulation of **e** neutrophil chemotaxis ($n = 10$) and **f, g** MΦ chemotaxis/chemokinesis, chemoattractive gradients, such as N-formylmethionyl-leucyl-phenylalanine (fMLP), monocyte chemotactic protein (MCP-1) and RGM-A, were established between a range of eight peripheral wells and a central cell loading well. PMN and M1 MΦ chemotaxis were evaluated using a Casy TT cell counter (Omni Life Science, Bremen, Germany) ($n = 18$). **f** Images represent the trafficking M1 MΦ (×100 magnification). **g** To determine the impact of RGM-A on macrophage chemokinesis, M1 MΦ were treated with RGM-A for 10 h and the migration toward the defined gradients was evaluated using Casy TT cell counter. **h** The rate of MΦ clearance of human apoptotic PMNs was assessed photometrically ($n = 11$) and by immunofluorescence (Scale bar: 20 μm). **i** mRNA expression of the ALX/FPR2 and GPR32 receptors ($n = 15$). The results are representative of 3–8 independent experiments and are expressed as the median ± 95% CI, one-way ANOVA with Bonferroni correction (**b–i**), unpaired two-tailed Student's *t*-test (**a**), *$P < 0.05$; **$P < 0.01$; ***$P < 0.001$, ****$P < 0.0001$

In the $RGM\text{-}A^{+/-}$ mice, we observed a significant increase in PMN recruitment to the peritoneum at 12 h, leading to a delayed resolution of inflammation, with a $Ri$ of ≈ 25 h (Fig. 2a). This response was accompanied by an enhancement in classical Ly6C$^{hi}$ monocytes and a decrease in non-classical Ly6C$^{low}$ monocytes, which, consequently, resulted in a reduction in efferocytosis (Fig. 2b). Furthermore, the pro-inflammatory cytokine levels were increased in the inflammatory exudates from the $RGM\text{-}A^{+/-}$ mice compared to their littermate controls (Fig. 2c).

**RGM-A enhances resolution and promotes tissue repair**. Next, we turned our attention to the potential effects of exogenous RGM-A. WT animals were injected with ZyA and subsequently with either vehicle or the RGM-A peptide, and lavages were collected at 4, 12, 24, and 48 h. Administration of RGM-A at the onset of inflammation significantly reduced the PMN influx into the peritoneum (Fig. 2d). This reduction was associated with a decrease in Ly6C$^{hi}$ monocytes and an increase in Ly6C$^{low}$ monocytes, which led to an enhancement in phagocytosis rate (Fig. 2d). When determining the resolution indices[19,20], we found that RGM-A shortened the PMN resolution interval (Ri) from 30 to 9 h (Fig. 2f). Moreover, RGM-A counter-regulated inflammation-initiating cytokines, such as Il-1β, IL-6, TNF-α, and KC, suggesting that RGM-A contributed to nonphlogistic cell recruitment (Fig. 2e). Because RGM-A demonstrated pro-resolving activity, we next sought to investigate its possible impact on tissue repair and regeneration. We performed immunohistochemical staining for proliferating-cell-nuclear antigen (PCNA) and found RGM-A to have pro-regenerative impact within the peritoneal tissue (Fig. 2g).

**RGM-A enhances pro-resolving lipid mediator biosynthesis**. During the inflammatory response, SPMs including lipoxins, resolvins, protectins and maresins exert pivotal biological effects by promoting resolution and restoring tissue homeostasis[20,21]. We monitored the temporal regulation of RGM-A during the initiation and in the resolution phase by evaluating the possible impact on the biosynthesis of SPMs during acute inflammation using LC-MS/MS-based profiling of murine peritonitis exudates. Exudate RGM-A was markedly increased at 4 h and subsequently decreased during the resolution phase (Fig. 2i). This temporal regulation of RGM-A occurred concomitantly with the induced biosynthesis of LXA$_4$, PDX (also known as 10S,17S-di HDHA) and Mar1 (Fig. 2h). Moreover, RGM-A significantly increased the arachidonic acid (AA)-derived products 15-HETE, 12-HETE, 5-HETE, as well as the eicosapentaenoic acid (EPA) product 15-

HEPE, 18-HEPE, and 14,15-diHETE (Fig. 2h). PGD$_2$ and PGE$_2$ were markedly elevated at 4 h following RGM-A administration compared to mice challenged with ZyA alone, whereas at 12 h (and 24 h), RGM-A decreased both factors, suggesting that RGM-A induced a mediator class switch (Fig. 2h, Supplementary Fig. 3, Supplementary Tables 1 and 2) from prostaglandins to the biosynthesis of anti-inflammatory and pro-resolving mediators within the inflammatory exudates[22,23]. Interestingly, we also observed activation in the CYP pathway from AA, where RGM-A significantly increased 14,15 EET, a lipid mediator which maintains anti-inflammatory and pro-resolution properties[24]. To corroborate these data, we determined the pro-resolving lipid mediators in RGM-A$^{+/-}$ mice. We found a marked increase of TXB$_2$ and LTB$_4$ and a strong reduction in the generation of pro-resolving mediators (SPMs) such as PDX and their pathway markers (Supplementary Fig. 3). The enzymes 5-LOX and 12/15-LOX are major players in the generation of pro-resolving mediators[25]. To provide additional mechanistic insights for the correlation between RGM-A and lipid mediator biosynthesis, we incubated peritoneal MΦ from WT or $12/15\text{-}LOX^{-/-}$ mice with RGM-A peptide and found reduced phagocytosis rate of fluorescently labeled ZyA particles after stimulation with RGM-A peptide (Supplementary Fig. 4a). In a second set of experiments we incubated human MΦ with RGM-A peptide and 5- and 12/15-LOX inhibitors cinnamyl-3,4-dihydroxy-α-cyanocinnamate (CDC) (Supplementary Fig. 4b) or baicalein (Supplementary Fig. 4c) and found significantly reduced MΦ phagocytosis rate, suggesting that the RGM-A-related effects in resolution are 5-LOX and 12/15 LOX dependent (Supplementary Fig. 4b-c). Using an additional lipopolysaccharide (LPS) induced peritonitis model the pro-resolving function of RGM-A could be verified (Supplementary Fig. 6a). We quantified leukocytes kinetics in various stages of inflammation observing strong effects on the resolution indices (Ri)[20], being reduced from 36 to 26 h (Supplementary Fig. 6b). Next, the lipid mediator analysis within the collected lavages induced the biosynthesis of SPMs and their pathway marker (Supplementary Fig. 6c-d).

**Adrenergic nerves regulate resolution of inflammation**. Recent research has identified the bidirectional communication between the immune system and the nervous system to be crucial in the pathophysiological mechanisms and homeostatic control[9]. However, recent studies and, finally, more detailed insights into neural immunoregulatory mechanisms show that the use of the sympathetic vs. parasympathetic model of neuron separation to describe inflammatory reflexes is restrictive or inaccurate[26].

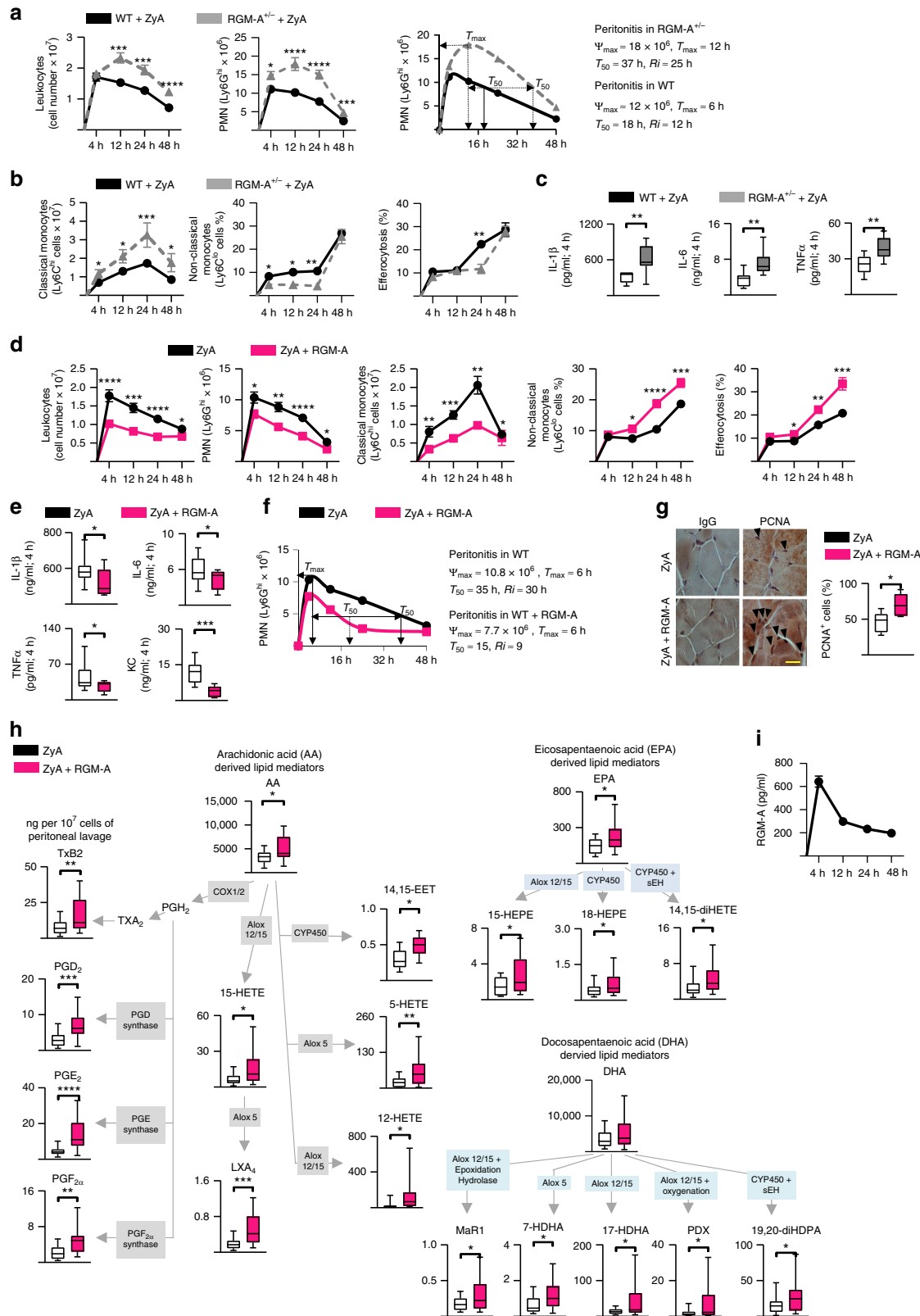

Therefore, it is of great importance to study closely the diversity of neurons in these complex circuits. Consequently, we sought to investigate a possible interaction between the sympathetic nervous system and RGM-A. For this, we first determined the expression of sympathetic α- and β-adrenergic receptors on human MΦ following the stimulation with TNF-α and/or RGM-A. Interestingly, we found RGM-A to strongly increase the $\beta_2AR$

mRNA expression, whereas $\beta_1AR$ and the α-adrenergic receptors $\alpha_{1A}AR$, $\alpha_{1D}AR$, $\alpha_{2A}AR$, and $\alpha_{2C}AR$ that are described to mediate pro-inflammatory responses during inflammatory events[27] were either suppressed or not significantly affected (Fig. 3a). Because RGM-A predominantly induces the expression of $\beta_2AR$ on MΦ compared with other subtypes of adrenergic receptors, we in turn stimulated human MΦ with TNF-α and/or the $\beta_2AR$ agonist

**Fig. 2** RGM-A$^{+/-}$ mice display impaired resolution of inflammation, while administered RGM-A enhances the resolution programs. Heterozygous RGM-A-deficient (RGM-A$^{+/-}$) mice and their littermate controls were injected with ZyA, and peritoneal lavages were collected at 4, 12, 24, and 48 h. **a** The total leukocytes were enumerated by light microscopy, and the PMNs were determined by flow cytometry. Resolution indices as defined in ref. [19] (WT $n = 20$, RGM-A$^{+/-}$ $n = 10$). **b** Classical and non-classical monocytes as well as the monocyte-derived macrophage efferocytosis was quantified by flow cytometry (WT $n = 20$, RGM-A$^{+/-}$ $n = 10$). **c** The IL-1β, IL-6, and TNF-α levels were measured in the peritoneal fluids by ELISA ($n = 9$). **d–i** WT animals were injected with ZyA and subsequently with vehicle or RGM-A, and lavages were collected at 4, 12, 24, and 48 h ($n = 20$). **d** The total leukocytes were quantified by light microscopy, and the PMNs, classical and non-classical monocytes as well as the macrophage efferocytosis were determined by flow cytometry. **e** The IL-1β, IL-6, TNF-α and KC levels were measured in the peritoneal fluids by ELISA ($n = 10$). **f** Resolution indices as defined in ref. [19]. **g** PCNA expression in the peritoneum (24 h after ZyA injection) using immunohistochemistry (magnification ×1000) ($n = 6$), Scale bar: 20 μm. **h** Levels of bioactive lipid mediators and precursors including the arachidonic acid (AA), docosahexanoic acid (DHA), eicosapentaenoic acid (EPA) and docosahexanoic acid (DHA) pathway were quantified by LC-MS/MS-based profiling in peritoneal fluids of WT animals that were treated with ZyA and RGM-A or vehicle for 4 h ($n = 20$). **i** The temporal regulation of RGM-A in ZyA-induced peritonitis exudates was assessed by ELISA. The results represent two independent experiments and are expressed as mean ± SEM (**a**, **b**, **d**, **i**) or median ± 95% CI (**c**, **e**, **g**, **h**), unpaired two-tailed Student's t-test, *$P < 0.05$; **$P < 0.01$; ***$P < 0.001$, ****$P < 0.0001$

formoterol and found that formoterol greatly increased the expression of RGM-A (Fig. 3b). Having demonstrated that RGM-A regulates the MΦ phenotype and function (Fig. 1), we next tested whether β$_2$-adrenergic signaling might play a role in the phenotypic differentiation or polarization of human MΦ. We observed the β$_2$AR agonist to be mainly involved in the phenotypic polarization toward the M2 MΦ as demonstrated by significant reduction in the levels of the M1 markers whereas the M2 markers and the cytokines *IL-10* and *TGF-ß* were significantly increased (Fig. 3c). To further investigate the role of sympathetic nervous system in the resolution of acute inflammation, we treated mice exposed to ZyA induced peritonitis with selective β$_2$AR agonist to mimic activation of adrenergic nerves. Initially, we performed immunochemistry staining of peritoneum to assess the impact of sympathetic adrenergic nerves on RGM-A expression. We identified increased expression of RGM-A within the neurofilament structures of the peritoneum following the stimulation with β$_2$AR agonist (Fig. 3d). Moreover, β$_2$AR agonist significantly enhanced the RGM-A mRNA expression within the peritoneum (Fig. 3e). Notably, when challenging mice to peritonitis and subsequently to RGM-A the β$_2$AR mRNA expression within the peritoneum was strongly enhanced (Fig. 3f), suggesting that there is an interplay between RGM-A and β$_2$-adrenergic signaling in murine peritonitis. To investigate the specificity of the reaction induced by sympathetic adrenergic nerves in peritoneal resolution tone WT animals were injected with ZyA and subsequently with either vehicle or β$_2$AR agonist, and lavages were collected at 4, 12, 24 and 48 h. Administration of β$_2$AR agonist significantly reduced the PMN influx into the peritoneum (Fig. 3g). This reduction was associated with a decrease in Ly6C$^{hi}$ monocytes and an increase in Ly6C$^{low}$ monocytes, which led to a reduction in phagocytosis (Fig. 3g). When quantifying the local kinetics of leukocyte migration[19], we found that β$_2$AR agonist shortened the PMN resolution interval (*Ri*) from 30 to 14 h (Fig. 3h). Using LC-MS-MS based lipid analysis we identified a significant increase of SPMs as well as their precursors and pathway markers. Particularly docosahexanoic acid- derived Mar1 and 17-HDHA were significantly increased following the administration of β$_2$AR agonist (Fig. 3i and Supplementary Table 1). We also found significantly increased levels of the arachidonic acid-derived products 5-HETE, 12-HETE, and 15-HETE, and the eicosapentaenoic acid-derived 14,15-diHETE (Fig. 3i). These modifications in lipid mediator profiles were accompanied with an increase in peritoneal norepinephrine concentrations—the principle neuromediator released from postganglionic neurons—4 h post ZyA (Fig. 3j). Surprisingly, when determining the peritoneal norepinephrine concentrations in *RGM-A$^{+/-}$* mice we found a significant decrease compared with littermate control (Fig. 3j), strongly implying that there is an interplay between RGM-A and

β$_2$-adrenergic signaling in ZyA induced peritonitis. Thus, these results indicate that adrenergic nerves promote key characteristics of resolution of acute inflammation.

**β$_2$-adrenergic signaling and RGM-A act synergistically**. Having demonstrated a mutual induction of both the β$_2$AR agonist and RGM-A in inflammatory in-vitro and peritonitis experiments, we next sought to investigate a potential interaction between β-adrenergic signaling and RGM-A and the underlying mechanisms by which resolution programs are affected. WT littermates were exposed to ZyA peritonitis and subsequent treatment with either RGM-A peptide or β$_2$AR agonist or both substances. We determined that RGM-A alone has a stronger impact on resolution processes than β$_2$AR agonist but when RGM-A and β$_2$AR agonist were co-administered, the synergistic effect was much more powerful as demonstrated key signs of resolution such as the greater decrease in leukocyte number and classical Ly6C$^{hi}$ monocytes and the greater increase in non-classical Ly6C$^{low}$ monocyte and in the phagocytosis rate (Fig. 4a). This synergistic impact of RGM-A and the β$_2$AR stimulation was also reflected in the generation of SPMs, including LXA$_4$, Mar1, 14, 15 EET, and their precursors and pathway markers (Fig. 4b and Supplementary Table 1). To obtain more mechanistic insight in the RGM-A–β$_2$AR axis we incubated peritoneal MΦ from *RGM-A$^{fl/fl}$/LysMcre-* and *RGM-A$^{fl/fl}$/LysMcre+* with a β$_2$AR agonist and determined the phagocytosis rate of fluorescently labeled ZyA particles. Our results showed that the phagocytotic activity of β$_2$AR agonist was significantly reduced in *RGM-A$^{fl/fl}$/LysMcre+* mice compared to the control group (Supplementary Fig. 4d). Next, we incubated human MΦ with a β$_2$AR agonist and 5- and 12/15-LOX inhibitor baicalein. The gathered data demonstrated a decrease in the phagocytosis rate compared to the control group (Supplementary Fig. 4e). Finally, we stimulated human MΦ with RGM-A peptide, β$_2$AR agonist and baicalein, and found that the phagocytosis rate was strongly reduced (Supplementary Fig. 4f). This finding suggests that RGM-A synergistically with formoterol is exposed to the negative influence of baicalein on the phagocytic rate, thereby strengthening our finding that RGM-A interacts with SPM biosynthesis and the adrenergic signaling. Both RGM-A and β$_2$AR agonist showed pro-resolving impact on acute inflammation. This effect is intensified by the synergistic effect of both substances (Fig. 4).

**Chemical sympathectomy impacts resolution programs**. To further underpin these synergistic effects of RGM-A and β$_2$-adrenergic signaling in resolution programs, we tested whether chemical sympathectomy would lead to perturbation of the pro-resolving tone. Firstly, we performed histological studies to

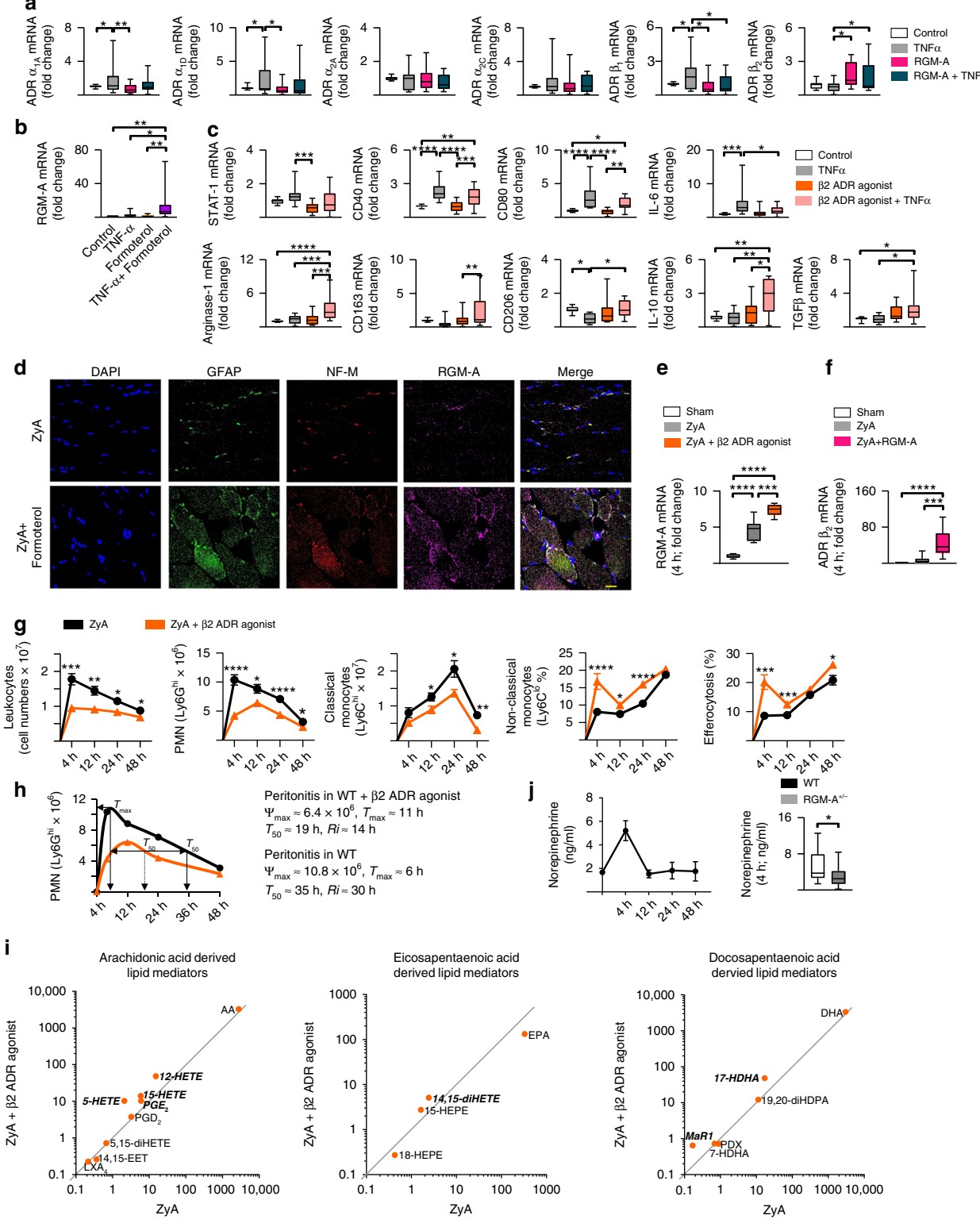

determine the RGM-A expression in 6-Hydroxydopamine hydrochloride (6-OHD) mice, in which we were able to detect a marked reduced expression of RGM-A in the neurofilament structures of the peritoneum (Supplementary Fig. 5). Then, we treated mice with 6-OHD to deplete peripheral adrenergic nerves 7 d before initiating inflammation. The results from this demonstrated that the administration of 6-OHD significantly lowered resolution as assessed by increased recruitment of PMN and classical monocytes followed by decreased non-classical

monocytes and finally lower clearance of apoptotic PMN and the generation of SPMs compared with vehicle control (Fig. 4c, d and Supplementary Table 3). Next, we investigated whether RGM-A could rescue this hyperinflammatory response. Indeed, administration of RGM-A rescued the excessive increased PMN infiltration and regulated the monocyte recruitment within the peritoneal exudates. In these exudates RGM-A also leveled out the SPMs generation as seen for LXA4, PDX, 14,15-EET and their pathway markers. (Fig. 4e and Supplementary Table 3).

**Fig. 3** Adrenergic nerves regulate resolution of inflammation. **a** Human M1 MΦ were stimulated with RGM-A and subsequently with TNF-α or vehicle for 24 h. The gene expression of the $\alpha_{1A}$, $\alpha_{1D}$, $\alpha_{2A}$, $\alpha_{2C}$, $\beta_1$, and $\beta_2$ adrenergic receptors (ADR) was quantified by RT-PCR. Human M1 MΦ were stimulated for 24 h with the $\beta_2$AR agonist and TNF-α or vehicle ($n = 15$). **b** The RGM-A mRNA was measured by RT-PCR ($n = 14$). **c** The gene expression of M1 polarization markers including *STAT-1*, *CD40*, *CD80* and *IL-6* as well as key genes of the M2 polarization such as *Arg1*, *CD163*, *CD206*, *IL-10*, and *TGFβ* were quantified by RT-PCR ($n = 14$). **d** WT animals were injected with ZyA and vehicle or $\beta_2$AR agonist for 4 h. The expression of RGM-A within the neurofilament structures of peritoneum analyzed by immunofluorescence (Scale bar: 20 µm). **e** The expression of *RGM-A* mRNA was analyzed in the peritoneum by RT-PCR ($n = 5$). **f** WT mice were treated with ZyA and RGM-A or vehicle for 4 h and the gene expression of $\beta_2$ ADR was measured in the peritoneum ($n = 7$). WT animals were injected with ZyA and with vehicle or $\beta_2$AR agonist, and lavages were collected at 4, 12, 24, and 48 h. **g** The total leukocytes were enumerated by light microscopy, and the PMNs by flow cytometry. The classical and non-classical monocytes as well as the MΦ phagocytosis rate were determined by flow cytometry ($n = 14$). **h** Resolution indices as defined in ref. [19]. **i** Levels of bioactive lipid mediators and precursors including the AA, EPA and DHA pathway were quantified by LC-MS/MS-based profiling in peritoneal fluids of WT animals that were treated with ZyA and $\beta_2$AR agonist or vehicle for 4 h. The data are shown as the geometric mean in ng per $10^7$ cells of peritoneal lavage and significant results are written in bold and italic ($n = 9$). **j** Norepinephrine was quantified in the peritoneal fluids of WT mice challenged to ZyA and *RGM-A*$^{+/-}$ mice and their littermates by ELISA ($n = 8$; $n = 15$ for 4 h time point). The results represent three independent experiments and are expressed as the median ± 95% CI (**a**–**f**, **j** (right)), geometric mean (**i**) and mean ± SEM (**g**, **j** (left)), one-way ANOVA with Bonferroni correction (**a**–**c**, **e**–**f**), unpaired two tailed Student's t-test (**g**, **i**, **j**), *$P <$ 0.05; **$P <$ 0.01; ***$P <$ 0.001, ****$P <$ 0.0001

**Monocyte/macrophage intracellular signaling**. Next we sought to gain further insight into mechanisms by which RGM-A and $\beta_2$-adrenergic nerves promote resolution programs. We focused on murine peritoneal MΦ that were collected 12 h after induction of peritonitis and treatment with either RGM-A or RGM-A and $\beta_2$AR agonist for analysis. Analysis of protein microarray data showed that RGM-A is involved in suppression of NF-κB activity, which is known to be a crucial transcriptional regulator of the M1 program[28] (Fig. 5a). Moreover, RGM-A regulates the m-TOR signaling pathway known to be an important driver in regulating MΦ metabolism and functional phenotype by activating RICTOR signaling to promote M2 activation[28] (Fig. 5b). Surprisingly, the MΦ treated with RGM-A and the $\beta_2$AR agonist additionally activated the PI3K/AKT pathway, which is important in confining pro-inflammatory responses stepping up anti-inflammatory responses and activating the monocytes/MΦ differentiation and polarization towards a pro-resolving phenotype (Fig. 5c, Supplementary Fig. 7 and Supplementary Data 1).

**Plasma RGM-A in critically ill children**. In order to show a translational significance, we investigated the association between RGM-A plasma levels from day of admission to and discharge from pediatric intensive care unit (PICU) and PRISM III score, intraabdominal hypertension (IAH) grade in a cohort of 109 critically ill children partly suffering from abdominal compartment syndrome (ACS)[29,30]. Medical and surgical patients ranging in age from newborn to <18 years were prospectively enrolled on admission in the PICU. Vital signs, drug administration, intra-abdominal pressure (IAP), fluid balances and other cardiorespiratory parameters were recorded continuously. Because of the heterogeneous severity of illness, subjects were divided into three test groups according to PRISM III-score, organ dysfunction and intraabdominal pressure (IAP) level. Patient demographic and clinical data are shown in Fig. 6a, b, Table 1 Supplementary Table 4. Our data revealed a significant change in plasma RGM-A levels in critically ill patients with IAH and with ACS during their PICU stay: Critically ill patients with IAH had 1.97-fold higher RGM-A plasma concentrations at day of discharge compared to the day of admission. In patients with a fulminant ACS, circulating RGM-A levels decreased by factor 0.55 from day of admission to discharge. When comparing plasma RGM-A levels with severity of illness, we found significant correlation with PRISM-III score and IAH grade. Notably, when divided into subgroups, a correlation of plasma RGM-A with PRISM-III score was only found in critical ill patients with ACS at day of admission (Table 2). At day of discharge we found significantly decreased levels of RGM-A, implying complicated recoveries with non-resolving outcomes. In all patients, we found a correlation between RGM-A plasma concentrations on the day of admission and the calculated average norepinephrine perfusion rate of PICU stay (24 h per kg body weight) (Fig. 6e). Interestingly, we did not observe such correlation at later time points or with other catecholamines than norepinephrine. These findings substantiate our in vivo data where RGM-A levels (with or without $\beta_2$ agonist stimulation) were markedly increased at the peak of inflammation in murine peritonitis fluids, and subsequently decreased, implying that RGM-A probably had a pro-resolving impact, particularly in the early resolution phase. Moreover, we compared conventional laboratory inflammatory markers with the previously mentioned descriptive, organ and outcome variables. There were comparably good correlations between plasma RGM-A (on day of admission) and CRP (rho- and P-values). Procalcitonin (PCT) to some extent even showed higher correlation coefficients with plasma RGM-A, but these results did not reach significance, probably due to the low N number of PCT samples (Supplementary Table 4). As expected, only some of the variables showed significant correlations with plasma RGM-A on day of discharge (Supplementary Table 4). To clarify whether an increased cell lysis might have caused an increase in RGM-A we analyzed the concentrations of lactate dehydrogenase (LDH). Thus, there was no difference in serum LDH between groups both on day of admission and at day of discharge, showing that the main source of RGM-A does not arise from cell lysis (Supplementary Fig. 8).

## Discussion

Acute inflammation is part of the protective response to infection, tissue stress and injury, representing a major challenge to homeostasis. It is commonly acknowledged that non-resolving inflammation displays an inherent risk for progression into chronic disease and organ failure[31,32]. The key issue with inflammation is not its frequent occurrence, but the failed resolution that can result in severe critical illness, as observed in pathologies such as peritonitis, respiratory distress syndrome or sepsis. Fundamental principles and steps in the inflammatory response and host defense start with a sequential release of chemical mediators, such as chemokines and lipid mediators, to induce PMN influx followed by monocyte-macrophage efferocytosis, leading to leukocyte clearance and resolution that ultimately enable homeostasis[21]. Hence, checkpoints exist early and late after an inflammatory response to control the inappropriate progression of inflammation[33]. The resolution process is concomitantly activated upon the initiation of inflammation and consists of a highly coordinated system that is mainly governed by endogenous specialized pro-resolving lipid mediators, such as lipoxins, resolvins, protectins, and maresins[34].

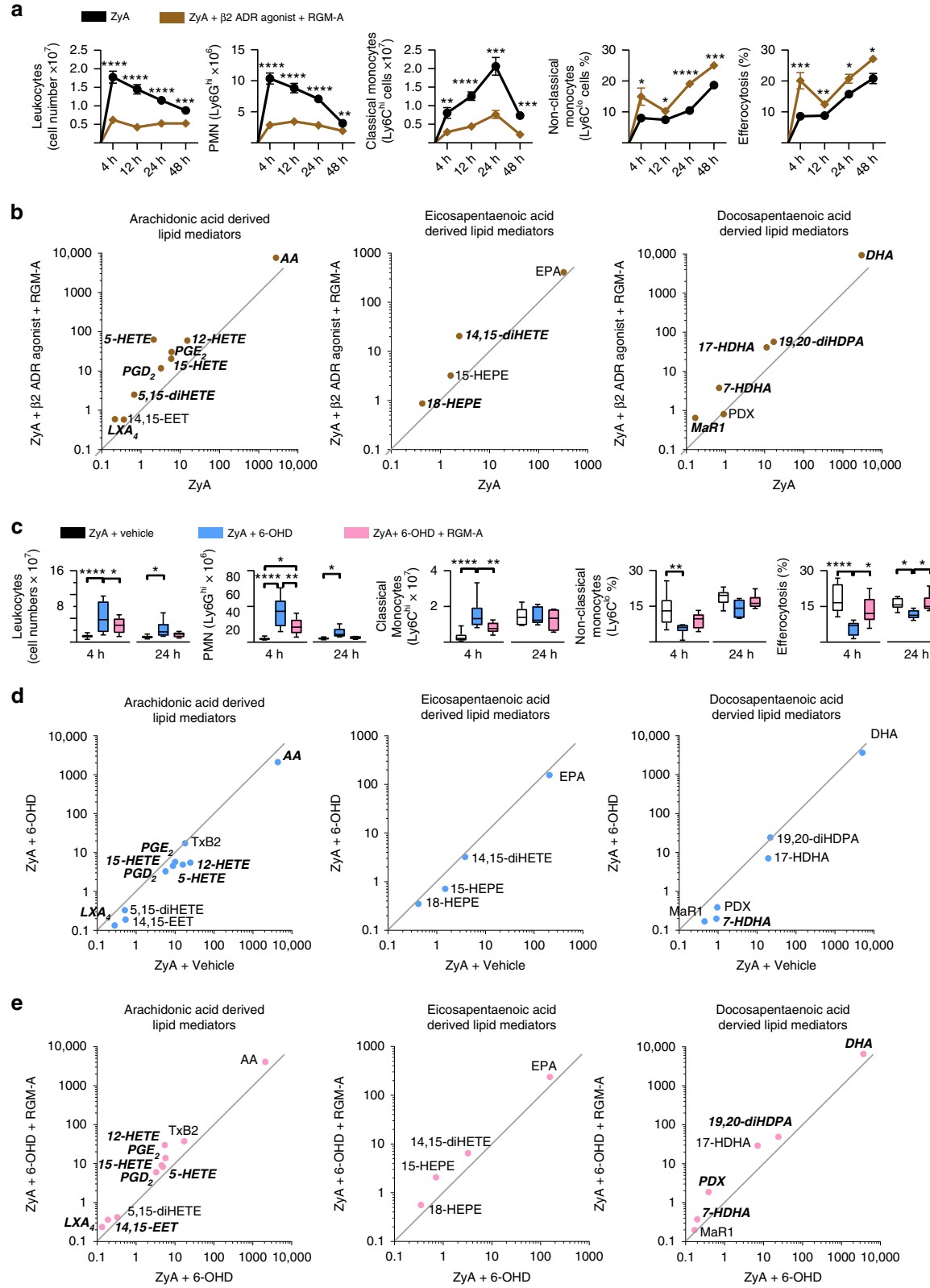

Numerous studies have illustrated a link between the immune and nervous systems identifying neuronal guidance proteins to be important players in immune function and chronic inflammation due to their chemoattractive and/or chemorepulsive capacity[7,35–38]. In the neuro-immune dialog new aspects of the neuronal reflex circuits have been characterized where the neuronal reflexes, sense peripheral inflammation and arrange inflammatory events within the initiation of inflammation[9,39]. Recently, we identified cholinergic nerve signaling to control the generation of immunoresolvents such as the neuronal guidance protein Netrin-1 and the SPMs during acute inflammation[10]. Based on these findings and because it is commonly acknowledged that endogenous anti-inflammation and pro-resolution are not equivalent processes, we opted to examine the role of sympathetic nervous system (SNS)

**Fig. 4** β2-adrenergic signaling and RGM-A synergistically activate inflammation-resolution programs. **a**, **b** WT animals were injected with ZyA and with vehicle or $β_2AR$ agonist and RGM-A, and lavages were collected at 4, 12, 24, and 48 h. **a** The total leukocytes were enumerated by light microscopy, and the PMNs, classical and non-classical monocytes as well as the MΦ efferocytosis by flow cytometry ($n = 12$). **b** Levels of bioactive lipid mediators and precursors including the AA, EPA and DHA pathway were quantified by LC-MS/MS-based profiling in peritoneal fluids of WT animals that were treated with ZyA and with RGM-A and Formoterol or vehicle for 4 h. Data are shown as the geometric mean in ng per $10^7$ cells of peritoneal lavage and significant results are written in bold and italic ($n = 10$). **c**, **d** To induce a chemical sympathectomy WT animals were treated with 6-Hydroxydopamine hydrochloride (6-OHD) or vehicle 7, 5, 3 d before the ZyA and vehicle or RGM-A injections. Lavages were collected at 4 ($n = 10$) and 24 h ($n = 7$) and **c** total leukocytes were enumerated by light microscopy. PMNs, classical and non-classical monocytes as well as the MΦ efferocytosis were analyzed by flow cytometry. **d** Levels of bioactive lipid mediators and precursors including the arachidonic acid (AA), eicosapentaenoic acid (EPA) and docosahexaenoic acid (DHA) pathway were quantified by LC-MS/MS-based profiling in murine peritonitis fluids that were treated with 6-OHD and vehicle or **e**) RGM-A for 4 h ($n = 10$). The results represent three independent experiments and are expressed as the mean ± SEM (**a**), geometric mean (**b**, **d**, **e**) and median ± 95% CI (**c**), unpaired two-tailed Student's $t$-test (**a**, **b**, **d**, **e**), one-way ANOVA with Bonferroni correction (**c**), *$P < 0.05$; **$P < 0.01$; ***$P < 0.001$, ****$P < 0.0001$

combined with the immunomodulatory actions of RGM-A in regulating resolution mechanism.

Given that recent studies indicate that specific cell morphologies and gene expression patterns mark differentiation into M1 or M2 phenotypes[14,15], we have shown that activation of monocytes with RGM-A induces differentiation to the M2 phenotype. Further data indicated $β_2AR$ agonist and RGM-A to play a direct role in the phenotypic polarization of MΦ. To further assess these pro-resolving actions, we determined the role of endogenous RGM-A in murine peritonitis. In WT littermates, the acute infiltration of leukocytes was initiated during the initial phase of inflammation, with maximal PMN infiltration at 4 h, followed by a decline, giving a resolution interval ($Ri$) of $\approx$ 12 h. In $RGM-A^{+/-}$ mice, we observed a significant increase in PMN recruitment to the peritoneum at 12 h, leading to a delayed resolution of inflammation, with an $Ri$ of $\approx$ 25 h. Exogenous RGM-A administered at the onset of inflammation significantly reduced the PMN influx into the peritoneum. When determining the resolution indices[19], we found that RGM-A shortened the PMN resolution interval ($Ri$) from 30 h to 9 h. Moreover, RGM-A counter-regulated inflammation-initiating cytokines, pointing to the contribution of RGM-A to nonphlogistic cell recruitment.

During the inflammatory response, SPMs exert pivotal biological effects by promoting resolution to return affected tissues to homeostasis[21]. We monitored the temporal regulation of RGM-A during the initiation and resolution phases and evaluated the possible impact on the biosynthesis of SPMs during acute inflammation by carrying out LC-MS/MS-based profiling of murine peritonitis exudates. The exudate RGM-A markedly enhanced at 4 h and subsequently decreased during the resolution phase. This temporal regulation of RGM-A was concomitant with the induced biosynthesis of $LXA_4$, PDX, and Mar1. $PGD_2$ and $PGE_2$ were markedly increased at 4 h after RGM-A administration, whereas at 12 h (and 24 h), RGM-A decreased the levels of both, suggesting that RGM-A induced a mediator class switch from prostaglandins to the biosynthesis of anti-inflammatory and pro-resolving mediators within the inflammatory exudates[22].

As mentioned above the nervous system is important to regulate immune homeostasis via neuronal reflexes[39]. Insight into neural immunoregulatory mechanisms indicates that the use of the sympathetic vs. parasympathetic model of neuron dissection to describe inflammatory reflexes is limiting[26]. Therefore, it is crucial to examine closely the variety of neurons in these complex circuits. We focused on a possible interaction between the catecholaminergic signaling and RGM-A and initially found RGM-A to strongly increase the $β_2AR$ mRNA expression, whereas $β_1AR$ and the α-adrenergic receptors $α_{1A}AR$, $α_{1D}AR$, $α_{2A}AR$, and $α_{2C}AR$ that are known to mediate pro-inflammatory responses during inflammatory processes[27] were either suppressed or not significantly affected. In ZyA-induced murine peritonitis, we identified enhanced expression of RGM-A within the neurofilament structures of the peritoneum following the stimulation with

$β_2AR$ agonist compared to vehicle control. In return, RGM-A increased the $β_2AR$ mRNA expression within the peritoneum when challenging mice to peritonitis, implying that there is an interaction between RGM-A and β-2 adrenergic signaling. When focusing specifically on the impact of adrenergic nerves on the characteristics of resolution of acute inflammation, we found that catecholaminergic signaling strongly promoted the key attributes of resolution, but not to the same extent as RGM-A. Interestingly, the synergistic effect of both substances displayed to be much more powerful as reflected in particular in the generation of SPMs. These synergistic effects were further substantiated by chemical sympathectomy, which resulted in disruption of pro-resolving tone, while treatment with RGM-A rescued the excessive increase in PMN infiltration and regulated monocyte recruitment within peritoneal lavage. Protein microarray data showed repression of the NF-κB, activation of RICTOR signaling and PI3K/AKT signaling in peritoneal monocytes after the stimulation with RGM-A and/or $β_2AR$ agonist.

It is obvious that non-resolving processes, such as those found in critically ill patients, can induce severe comorbidities that pose major threat to global health. In fact, clinical trials may be the most effective way to examine the usefulness of novel predictive indications. Therefore, in a cohort of ICU critically ill pediatric patients we found significantly increased RGM-A blood plasma levels in the IAH patient cohort, whereas in the ACS cohort including patients with sustained non-resolving processes and finally high mortality rate, RGM-A levels were significantly reduced, suggesting that RGM-A may predict clinical outcomes. Interestingly, we found a correlation between RGM-A and norepinephrine substitution—and not with other catecholamines—only initially on the day of admission to the intensive care unit, and not later. These findings complement our in vivo data, where murine peritonitis exudate displayed markedly increased RGM-A levels at the peak of inflammation and subsequently decreased, suggesting that RGM-A induced its pro-resolving effects in the early resolution phase.

Results of the present experiments identify a new aspect of the neural-reflex circuit that controls key processes in the resolution of acute inflammation and promotes tissue repair and regeneration and may provide potential new insights to improve our understanding of the inflammation/resolution process in innervated organs, as well as potential therapeutic value for the treatment of acute inflammatory conditions.

## Methods

**Human leukocyte isolation and macrophage (MΦ) differentiation and polarization.** Human peripheral blood monocytes (PBMCs) were isolated from healthy volunteers or human leukapheresis collars from the Blood Bank of Eberhard Karls University of Tübingen and cultured in RPMI 1640 medium with 10 ng/ml human recombinant GM-CSF (Macs Milteny Bergisch Gladbach, Germany), 100 ng/ml M-CSF (Macs Milteny) or 250 ng of RGM-A peptide at 37 °C for 7 d. For polarization, M1 (cultured with GM-CSF for 7 d) or M2 (cultured with M-CSF for 7 d) macrophages were stimulated with 100 ng of TNF-α (Promokine, Heidelberg,

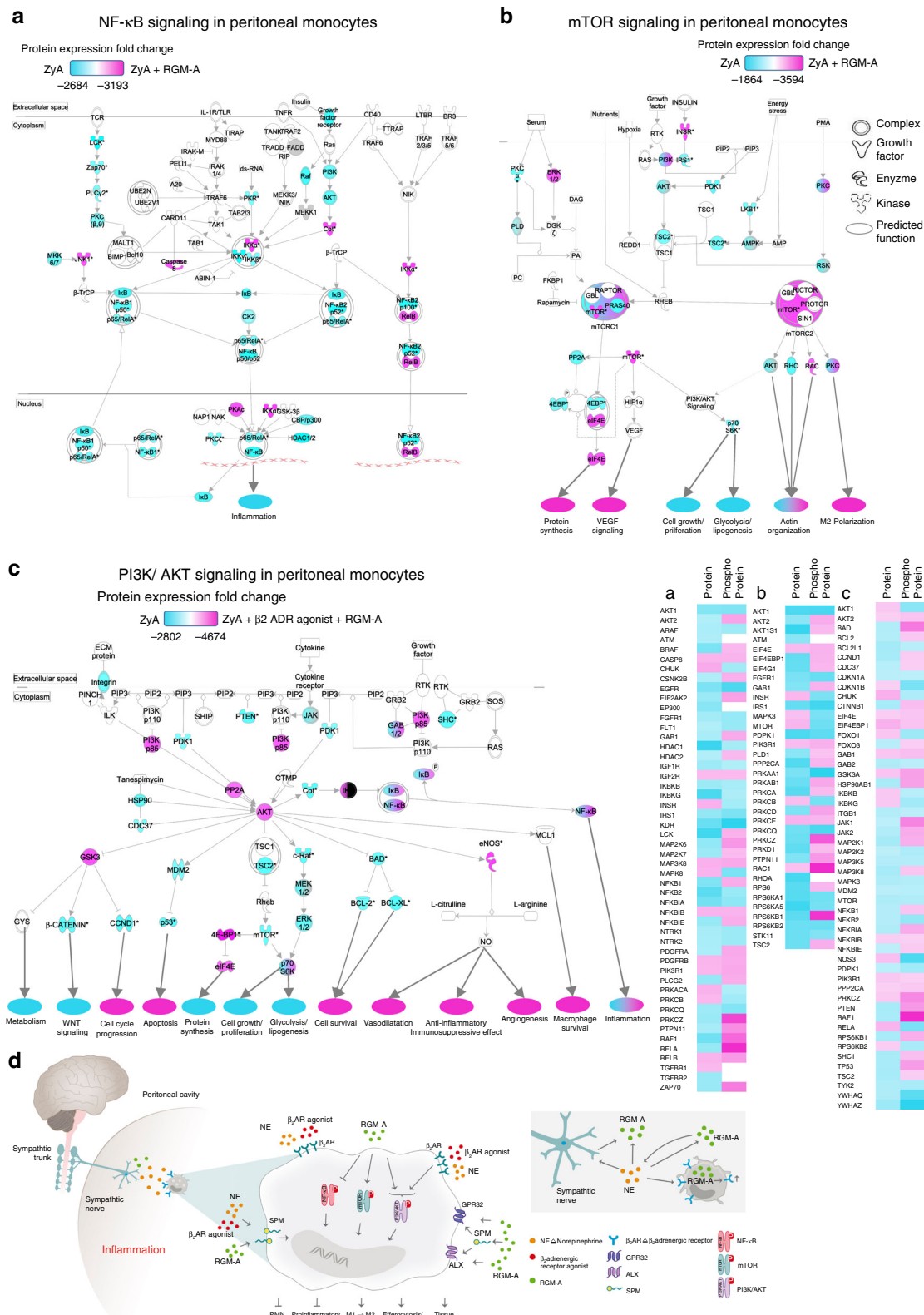

**Fig. 5** RGM-A-dependent monocyte intracellular signaling. WT animals were exposed to ZyA induced peritonitis and treated with with RGM-A and/or β2AR agonist or vehicle and peritoneal monocytes were collected 12 h after ZyA injection. **a** The NF-κB, **b** the mTOR, and **c** the PI3K/Akt signaling pathways were assessed in peritoneal monocytes by using a protein microarray. Samples were pooled from 3 to 4 mice in each group for each experiment. **d** Schematic model of the interaction and synergistic effect of sympathetic nervous system and RGM-A

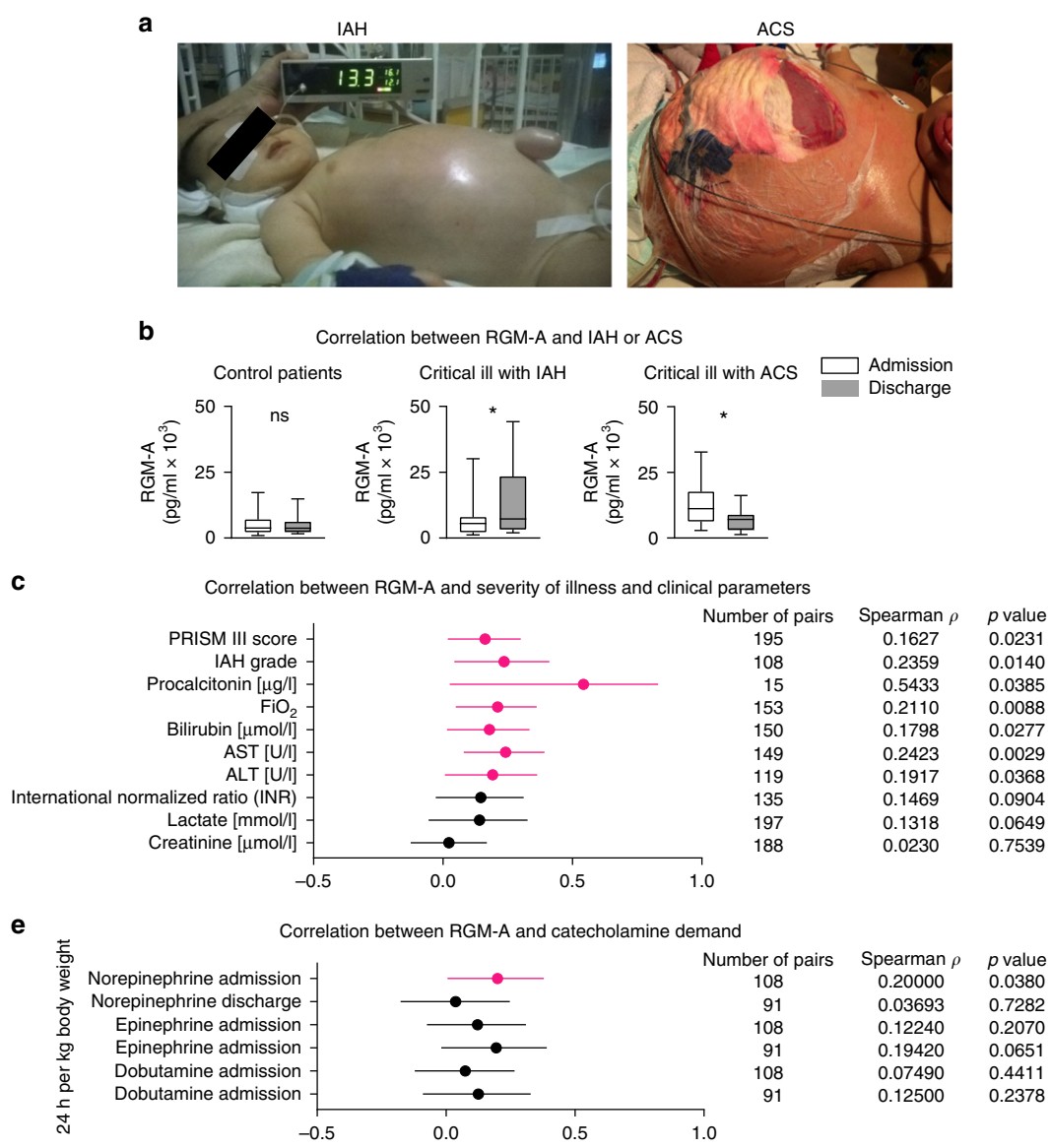

**Fig. 6** RGM-A in pediatric ICU patients with abdominal compartment syndrome. Plasma samples from 109 ICU children with and without abdominal compartment syndrome (ACS) were collected within 24 h after admission to and at day of discharge from the Pediatric Intensive Care Unit (PICU) of Hannover Medical School (MHH, Germany). **a** Representative images displaying critical ill children with IAH (conservative therapy management) (left) and ACS (after decompressive laparotomy with the establishment of an open abdomen / laparostoma to reduce intra-abdominal pressure and the associated tissue and organ impairments) (right). Despite surgical decompression, the child from the ACS cohort died of uncontrollable multiple organ failure. **b** Overview of ICU patient characteristics within the control group (Prism-III-score < 8), critically ill children with IAH (IAP > 10 mmHg) but without ACS and critically ill children meeting the WSACS definitions[29] for ACS. **c** Correlation between RGM-A and clinical parameters of all PICU patients at day of admission enrolled. **d** Correlation between RGM-A and risk of mortality at day of admission in subgroups. **e** Correlation between RGM-A at day of admission and discharge and the average infusion rate of the catecholamines Epinephrine, Norepinephrine and Dobutamine. The Spearman's rank correlation coefficient Rho and the corresponding 95% CI interval are shown. Results are displayed as median ± 95% CI; non-parametric Mann-Whitney test; correlation was tested using Spearman's rank correlation test; *$P < 0.05$

Germany) ± 250 ng of RGM-A peptide ± 100 nM of formoterol (Sigma-Aldrich) for 24 h, and then transcriptional analysis was performed.

**Transcriptional analysis**. The transcriptional analysis of human *RGM-A* mRNA expression was performed using the sense primer 5′-AAC CAG CAG ATC GAC TTC CAG-3′ and antisense primer 5′-ACG GCT GTC TCG TAT GGG A-3′. Human *18S* expression was evaluated with the sense primer 5′-GTA ACC CGT TGA ACC CCA TT-3′ and antisense primer 5′-CCA TCC AAT CGG TAG TAG CG-3′. The following primers were used to determine the macrophage phenotype: *CD80*: 5′-AGC CTC ACC TCT CCT GGT TG-3′, 5′-TGG GGC AAA GCA GTA GGT CA-3′; *STAT-1*: 5′-ATC AGG CTC AGT CGG CGG ATA-3′, 5′-TGG TCT CGT GTT CTC TGT TCT-3′; *CD40* 5′-ACT GAA ACG AAA TGC CTT CCT-3′, 5′-CCT CAC TCG TAC AGT GCC A-3′; *Arg1*: 5′- TGG ACA GAC TAG GAA

TTG GCA-3′, 5′-CCA GTC CGT CAA CAT CAA AAC T-3′; *CD163*: 5′-ACA ACA GGT CGC TCA TCC C-3′, 5′-GTG TGG CTC AGA ATG GCC T − 3′; *CD206*: 5′- CCC TCA GAA AGT GAT GTG CCT-3′, 5′- TCT CCA CGA AGC CAT TTG GT-3′; *IL-1β*: 5′-GAC CAC CAC TAC AGC AAG GG-3′, 5′-ATC GTG CAC ATA AGC CTC GT-3′; *IL-6*: 5′-CAC CAG GCA AGT CTC CTC AT-3′, 5′-GAC AGC CAC TCA CCT CTT CA-3′; *IL-10*: 5′-AAT CGA TGA CAG CGC CGT AG-3′, 5′-GGT TGC CAA GCC TTG TCT GA-3′ and *TGFβ*: 5′-TGG TGG AAA CCC ACA ACG AA-3′, 5′-GAA GTT GGC ATG GTA GCC CT-3′.The transcriptional analysis of the G protein-coupled receptors (GPCRs) such as ALX/FPR2 and GPR32, which have been shown to mediate pro-resolving actions were performed using the following primers: *ALX/FPR2*: 5′-TGT TCT GCG GAT CCT CCC ATT-3′, 5′-CTC CCA TGG CCA TGG AGA CA-3′. *GPR32*: 5′-GGG CCT GCA AAC TCT ACA-3′, 5′-GGA GGC AGT TAC TGG CAA-3′. The transcriptional analysis of the adrenergic receptors were performed using the following

**Table 1 Patient statistics**

| | Number of patients | Male sex | Prism III score (median, min–max admission) | Age in month (median, min–max) | Days of PICU stay (median, min–max) | | Post surgery | Hemic-oncolog. | Cardiology/ pulmology | Sepsis | Post neuro surgery |
|---|---|---|---|---|---|---|---|---|---|---|---|
| PICU control patients | 58 | 41 (71%) | 8.0 (0–18) | 15.0 (0–215) | 3.0 (0–70) | Primary reason for admission on PICU | 10 | 3 | 33 | 0 | 12 |
| Critically ill with IAH | 32 | 19 (59%) | 13.0 (3–31) | 29.0 (0–199) | 6.0 (0–28) | | 12 | 0 | 17 | 0 | 3 |
| Critically ill with ACS | 19 | 9 (47%) | 18.0 (9–35) | 47.0 (0–203) | 13.0 (1–332) | | 12 | 2 | 4 | 1 | 0 |

**Table 2 Correlation between RGM-A and risk of mortality in subgroups**

| PRISM III Day of admission | RGM-A vs. control patients | RGM-A vs. Critically ill with IAH | RGM-A vs. Critically ill with ACS |
|---|---|---|---|
| Spearman $\rho$ | 0.09758 | −0.173 | 0.3623 |
| P value (two-tailed) | 0.3268 | 0.1788 | 0.0491 |
| Number of pairs | 103 | 62 | 30 |

primers: *ADR $\alpha_{1A}$*: 5′-CGC TAC CCA ACC ATC GTC A-3′, 5′-GAA CAG GGG TCC AAT GGA TAT G-3′, *ADR $\alpha_{1D}$*: 5′-CCA AGC TGT GCA AAC TGT CC-3′, 5′-TCT GTT GCG GGC CTT TAG TT-3′, *ADR $\alpha_{2A}$*: 5′-TCT TCA CCT ACA CGC TCA CG-3′, 5′-TGC AGT AGC CGA ACC AGA AG-3′, *ADR $\alpha_{2C}$*: 5′-CTA CTG GTA CTT CGG GCA GG-3′, 5′-ATG GCA CAC AGA TGC ACG AT-3′, *ADR $\beta_1$*: 5′-GAA GCC CAC AAT CCT CGT CT-3′, 5′-CGG TCC GTG GCT TTT CTC TT-3′, *ADR $\beta_2$*: 5′-ATG GGC ACT TTC ACC CTC TG-3′, 5′-GCT CCG GCA GTA GAT AAG GG-3′.

The transcriptional analysis of murine RGM-A and $\beta_2$ ADR expression was performed using the following primers: *mu RGM-A*: 5′-CTT CCC CGC AGC CAT CT-3′, 5′-CCT CTA TGC CAT GGA CAG CC-3′. *mu GAPDH*: 5′-ACA TCA AGA AGG TGG TGA AGC-3′, 5′-AAG GTG GAA GAG TGG GAG TG-3′; *mu ADR $\beta_2$*: 5′-AAT AGC AAC GGC AGA ACG GA-3′, 5′-TCA ACG CTA AGG CTA GGC AC-3′ and *mu 18S*: 5′-GTA ACC CGT TGA ACC CCA TT-3′, 5′-CCA TCC AAT CGG TAG TAG CG-3′.

**RGM-A peptide**. The RGM-A peptide -KYIGTTIVVRQVGRYLTFA- is composed of 19 amino acids spanning from position 275 to 293 from murine RGM-A (uniProt number q6pcx7), and this region is evolutionary conserved in humans (uniProt number Q96B86). The RGM-A peptide with an average mass of 2185.6 Daltons and theoretical isoelectric point of 10.28 was delivered lyophilized (Think peptides) and reconstituted with ultra pure water to a concentration of 1 µg/µl. This selected RGM-A region binds specifically neogenin and has been identified and tested by Itokazu et al[40].

**PMN and macrophage chemotaxis and chemokinesis**. Isolated human PMNs were prepared from the peripheral blood of healthy volunteers by gradient centrifugation. MΦ were differentiated from peripheral blood monocytes (as described above). The cells were stained with rhodamine-6G (Sigma-Aldrich) according to the manufacturer's instructions. The micro fluidic devices were printed using an S30L DLP printer (Rapidshape, Heimsheim, Germany) with photoresist MP300 (Rapidshape, Heimsheim, Germany). The device was constructed using Netfabb Professional 5.2 (Netfabb, Lupburg, Germany). Chemoattractant gradients, such as N-formylmethionyl-leucyl-phenylalanine (fMLP), monocyte chemotactic protein (MCP-1) and RGM-A (±MCP-1), were established between a range of eight peripheral chambers and a central cell loading well. The control chambers were loaded with RPMI. The cells were suspended at a density of $5 \times 10^4$ cells in 10 µl of total volume and placed into the central loading chamber. The cells were then incubated at 37 °C for 2 h (PMN) or 8 h (MΦ) prior to imaging to allow the chemotactic gradient to be generated. To determine the MΦ chemokinesis, MΦ were treated with RGM-A at 37 °C for 10 h prior to imaging. Cell migration was monitored using a LSM 510 Meta fluorescence microscope (Carl Zeiss, Jena, Germany) and enumerated with a Casy TT cell counter (Omni Life Science, Bremen, Germany). A detailed description of the chemotaxis and chemokine assessments is presented in the SI text.

**Human MΦ efferocytosis**. To prepare apoptotic PMNs, human PMNs obtained from peripheral blood were isolated and labeled with carboxyfluorescein diacetate (10 µM, 30 min at 37 °C; Molecular Probes) and allowed to undergo apoptosis in serum-free RPMI 1640 medium for 16–18 h. GM-CSF-differentiated

MΦ ($0.1 \times 10^6$ cells/well) were then incubated with human RGM-A peptide. Apoptotic PMNs were added at a 1:3 ratio (MΦ:PMN) and incubated at 37 °C for 60 min to allow phagocytosis. Fluorescence was determined using a fluorescent plate reader (Tecan, Männedorf, Switzerland). In a separate experiment, cells were stimulated with the LOX-inhibitors baicalein (Sigma Aldrich; #465119) and cinnamyl-3,4-dihydroxy-α-cyanocinnamate (CDC; Abcam; #ab141560). For immunofluorescence, human MΦ efferocytosis of fluorescently labeled ZyA particles in cells that were stimulated with RGM-A or vehicle was analyzed. DAPI (4′,6-diamidino-2-phenylindole; Invitrogen, #P36931) was employed for nuclear counterstaining

**Murine MΦ phagocytosis**. To prepare murine MΦs, mice were euthanized and peritoneal lavages collected and plated on 48-well plates for 1 h in phosphate-buffered saline with calcium and magnesium to allow adherence. Subsequently, cells were stimulated with RGM-A peptide or vehicle and fluorescently labeled ZyA particles were added and incubated at 37 °C for 60 min to allow phagocytosis. In a separate experiment, cells were stimulated with the LOX-inhibitors baicalein (Sigma Aldrich) and cinnamyl-3,4-dihydroxy-α-cyanocinnamate (CDC; Abcam). Fluorescence was determined using a fluorescent plate reader (Tecan, Männedorf, Switzerland).

**Animals**. All animal experiments were carried out according to the procedures approved by the Institutional Review Board and the Regierungspräsidium Tübingen and comply with all relevant ethical regulations regarding animal research. *RGM-A$^{+/-}$* and littermate control mice were bred and genotyped using the following primers: *wild type*; 5′- CAG GTA GGC ACA ACT CCT TGG TGG-3′, 5′- TTA GCA CGT CT G AGC CTG TGT CCG-3′; *knock-out*: 5′- TGC GAA GTG GAC CTG GGA CCG CG-3′, 5′- CAT CCA ACA AGG CTC CAC TGG AAG G-3′[7]. Floxed RGM-A mice were bred with LysM-Cre transgenic mice (The Jackson Laboratory) to generate myeloid cell lineage-specific knockout animals. LysM-Cre– floxed littermates were used as their control. 12/15-LOX-deficient mice (*12/15-LOX$^{-/-}$*; The Jackson Laboratory, # 002778) and littermate control mice were bred and genotyped using following primers: mutant: 5′- GGG AGG ATT GGG AAG ACA AT-3′, common: 5′-GGG TGC CTG AAG AGG TAC AG-3′, wild type: 5′- CCA TAG ACG AGA CCA GCA CA-3′[41].

**Murine model of ZyA-induced peritonitis**. All animal protocols were performed in accordance with the regulations of the Regierungspräsidium Tübingen and the local ethics committee. The mice were intraperitoneally (i.p.) injected with 1 ml of zymosan A (ZyA: 1 mg/ml; Sigma-Aldrich,) and subsequently with either vehicle or 0.5 µg of RGM-A peptide in a total volume of 150 µl. The $\beta_2$ agonist Formoterol (Sigma-Aldrich, 50 µg/kg body weight, #F9552) was injected i.p. together with ZyA and RGM-A and then every 12 h. To induce a chemical sympathectomy 6-Hydroxydopamine hydrochloride (6-OHD; Santa Cruz, #sc-203482, 100 mg/kg body weight with 0.1% Ascorbic acid in PBS) or vehicle were administered i.p. 7, 5, 3 d prior to ZyA and vehicle or RGM-A injections Peritoneal fluids and tissues were obtained at 4, 12, 24, and 48 h and prepared as previously described[10]. The collected exudates were washed, suspended in Hanks' balanced salt solution and counted, and Cytospin samples were prepared.

**Murine model of LPS-induced peritonitis**. All animal protocols were performed in accordance with the regulations of the Regierungspräsidium Tübingen and the local ethics committee. C57BL/6 mice were injected intraperitoneally (i.p.) with 0.2 ml of Lipopolysaccharide (LPS: 0.5 mg/ml; Sigma-Aldrich) and subsequently with either vehicle or 0.5 µg of RGM-A peptide in a total volume of 150 µl. Peritoneal fluids and tissues were collected at 4, 12, 24, and 48 h.

**Differential leukocyte counts, flow cytometry analysis, and ELISA**. Exudate cells from the murine peritonitis model were prepared to determine the cellular composition. The cells were blocked with mouse anti-CD16/CD32 (BioLegend, #101320, 1:50) antibodies for 10 min at room temperature and then stained with anti-mouse APC-Ly6G (BioLegend, # 127614, 1:250), anti-mouse e450-F4/80 (eBioscience, # 48-4801-82, 1:100) (and anti-mouse FITC-Ly6C (BioLegend, #128006, 1:250) antibodies for 30 min at 4 °C. To analyze the MΦ phagocytosis of apoptotic PMNs in vivo, the cells were permeabilized and then stained with anti-mouse PerCP-Cy5.5-conjugated anti-Ly6G (BioLegend, #127616, 15:1000) for 30 min at 4 °C. The cells were analyzed by flow cytometry (BD FACSCanto II). All FACS gating strategies are shown in Supplementary Fig. 9. Cytokines, RGM-A and Norepinephrine were measured in the murine peritoneal exudates using standard ELISA (R&D systems, LDN immunoassays).

**Cytology, immunofluorescent, and immunohistochemical staining**. Formalin fixed and paraffin-embedded (FFPE) peritoneal tissues were stained with mouse anti-GFAP A488-labeled (Biolegend, # 644704, 1:250), rabbit anti-tyrosine hydroxylase (abcam, #ab112, 1:1000), goat anti-NF-M (SantaCruz, #sc-16143, 1:50), rabbit anti-RGM-A (Abcam, #ab26287 1:50) or goat anti-RGM-A (Santa-Cruz, # sc-46481, 1:50) and IgG isotype control antibodies (Santa Cruz Biotechnology) as negative controls. Alexa Fluor 594-conjugated donkey anti-goat (Invitrogen, #A-11058, 1:100), Alexa Fluor 546-conjugated goat anti-rabbit (Invitrogen, #A-11010, 1:100) and Alexa Fluor 647-conjugated goat anti-rabbit (Invitrogen, #A-21244, 1:100) were used as secondary antibodies. Immunofluorescent pictures were acquired using a confocal microscope (LSM 510 Meta fluorescence microscope, Carl Zeiss) and ZEN software (Carl Zeiss). To perform immunohistochemical staining for PCNA, FFPE peritoneal tissues were stained with an anti-PCNA antibody (Santa Cruz Biotechnology) using a Vectastain ABC Kit (Vector Labs) and DAB peroxidase substrate (Sigma-Aldrich) according to the manufacturers' instructions. As secondary antibody a biotin-conjugated rabbit-anti-mouse antibody (Jackson ImmunoResearch) was used. The sections were then counterstained with hematoxylin.

**Antibody array for protein expression**. Peritoneal monocytes/macrophages from WT mice were used following 12 h of ZyA-induced peritonitis. Protein and phosphorylation (Phospho Explorer Antibody Array, FullMoonBioscience, #PEX100) profiling of peritoneal monocytes (pooled lavages from 4 mice per condition) was carried out according to the manufacturer's instructions. The images were acquired by the manufacturer. For each antibody, the average signal intensity of two replicates was normalized to the median signal of all antibodies on the array. The presented fold change represents the ratio of the normalized signal from WT mice challenged with either ZyA and RGM-A or in combination with Formoterol compared with WT challenged with ZyA alone. GAPDH and beta actin were used as housekeeping proteins. Data analysis was performed with IPA software (Qiagen). Pathways were substantiated and updated with recent literature, KEGG database (HSA 04150; HSA 04064; HSA 04151) and Reactome database (R-HSA-165159, R-HSA-5676590, R-HAS-198203).

**Pediatric ICU patient samples with and without abdominal compartment syndrome**. Plasma samples were taken from 109 pediatric ICU patients from the Pediatric Intensive Care Unit (PICU) of Hannover Medical School (MHH, Germany) within 24 h after admission and at day of discharge. Critically ill children between 0 and 18 years of age were enrolled between January and August 2015 after obtaining informed written consent from the parents or guardians of each child. The study was approved by the Local Ethics Committee (MHH-No. 6677) and internationally registered (WHO-ICTRP DRKS00006556). The 2013 WSACS definitions[29] (with respect to IAP and ACS; www.wsacs.org) were used to define the abdominal compartment syndrome (ACS). The severity of illness in the ICU children was measured using PRISM-III-scoring[30]. Vital and cardiorespiratory parameters (including ventilation times), drug administration, intra-abdominal pressures (measured via gastric Spiegelberg® monitoring system[42],) and fluid balances were recorded continuously via the digital patient data management system (mlife, mediside) A RGM-A ELISA (R&D Systems, #DY2459-05) was performed according to manufacturer's instructions.

**Ethics statement**. Ethical approval was obtained by the Ethics Review Committee of the Faculty of Medicine, Eberhard Karls University Tübingen (Approval number 351/2013BO2) and Hannover Medical School (MHH). Informed consent was obtained. We have complied with all relevant ethical regulations in carrying out this study.

**LC-MS/MS**. Peritoneal lavage samples were spiked with 4 µl of an internal standard solution (containing PGE4-d4, LTB4-d4 15-HETE-d8 and DHA-d6 at a concentration of 50 ng/ml in methanol). The samples were transferred to a 12-ml glass vial, and 1.75 ml of methanol was added. The samples were centrifuged at 4,000 rpm for 5 min at 6 °C, and the supernatant was transferred to a fresh 12-ml glass vial. The pellet was re-extracted with 500 µl of methanol and centrifuged as described above, and the organic extracts were combined. The methanol was partially removed under a gentle stream of nitrogen at 40 °C for 30 min. The remaining methanolic extract (approximately 1.5 ml) was diluted with 8 ml of water, and 20 µl of 6 M HCl was added. The prepared samples were cleaned via solid phase extraction (SPE) (SepPak C18 200 mg, Waters, MA, USA). The samples were loaded onto preconditioned SPE cartridges (2 ml methanol, followed by 2 ml water), the cartridges were washed with 3 ml of water followed by 3 ml of n-hexane, and then the samples were eluted with 3 ml of methylformate. The eluate was dried under a gentle stream of nitrogen, reconstituted in 200 µl of 40% methanol, and injected.

LC-MS/MS analysis was performed as described below. Briefly, a QTrap 6500 mass spectrometer operating in negative ESI mode (Sciex, Nieuwerkerk aan den Ijssel, The Netherlands) was coupled to an LC system employing two LC-30AD pumps, a SIL-30AC autosampler, and a CTO-20AC column oven (Shimadzu,'s-Hertogenbosch, The Netherlands). A 1.7 µm Kinetex C18 50 × 2.1 mm column protected with a C8 precolumn (Phenomenex, Utrecht, The Netherlands) was used, and the column was maintained at 50 °C. A binary gradient of water (A) and MeOH (B) containing 0.01% acetic acid was generated as follows: 0 min 30% B, held for 1 min, then ramped to 45% B at 1.1 min, 53.5% B at 2 min, 55.5% B at 4 min, 90% B at 7 min, and 100% B at 7.1 min, and held for 1.9 min. The injection volume was 40 µl, and the flow rate was 400 µl/min. For analyte identification, the mass transition used for each analyte was combined with its relative retention time (RRT). The calibration lines constructed with standard material for each analyte were used for quantification, and only peaks with a signal to noise (S/N) ratio > 10 were quantified[43,44].

**Data analysis**. The data were compared by one-way ANOVA with Bonferroni correction or unpaired two tailed Student's $t$-test as appropriate unless otherwise stated and indicated in the figure legends using GraphPad Prism 8.0. For statistical tests *$P < 0.05$, **$P < 0.01$, ***$P < 0.001$, ****$P < 0.0001$.

**Reporting summary**. Further information on experimental design is available in the Nature Research Reporting Summary linked to this article.

## Data availability

The authors declare that all data supporting the findings of this study are available within the article and its Supplementary Information Files or from the corresponding author upon reasonable request. The antibody array data generated and analyzed during the current study are available in the NCBI's Gene Expression Omnibus database[45] through GEO Series accession number GSE122569.

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

## Acknowledgements

This work was supported by a grant from the Interdisziplinäres Zentrum für Klinische Forschung (IZKF) 2110-0-0 and two grants from the Deutsche Forschungsgemeinschaft (DFG-MI 1506/4-1, DFG-MI 1506/5-1) to V.M.; by the DFG CRC/TR 240 funded by the Deutsche Forschungsgemeinschaft (DFG, German Research Foundation) – Projektnummer 374031971 – TRR 240 (to V.M.); by a IZKF – fortüne grant 2377-0-0 to A.K. and by a IZKF – fortüne grant 2299-0-0 to M.S. The collection of clinical data was made possible by an intramural sponsorship of T.K. through the MHH (Young Academy – Clinical Scientist Program). We thank Alice Mager, Alice Bernard, Urs Knausberg and Marieke Heijink for their technical support. We thank Tiago Granja for composing and ordering the RGM-A peptide. We thank Robert Steiner and Andreas Schultheiss (Rapid Shape GmbH; generative 3D rapid prototyping and manufacturing; Heimsheim; Germany) for manufacturing the microfluidic devices and their technical support.

## Author contributions

A.K., M.S. and V.G. performed the experiments, collected, and analyzed the data. M.G. performed the targeted lipidomic and lipid mediator analysis studies. T.K. and T.S. designed and performed the patient data analysis. T.K., T.S. and G.H. performed patient data analysis. All authors contributed to manuscript and figure preparation. V.M. carried out overall experimental design and conceived of the overall research project. V.M. wrote the manuscript.

## Additional information

**Competing interests:** The authors declare no competing interests.

**Journal Peer Review Information**: *Nature Communications* thanks the anonymous reviewers for their contribution to the peer review of this work. Peer reviewer reports are available.

