## [Peer Review File · Nature Communications]

Reviewers' comments:

Reviewer #1 (M1/M2 macrophage, metabolism)(Remarks to the Author):

The current manuscript examines the effect of RGM-A on macrophage inflammatory response and resolution. Using human monocytes and zymosan A-induced peritonitis in mice, the authors demonstrated that RGM-A suppresses infiltration of inflammatory macrophages, which is associated with increased levels of pro-resolving lipid mediators. In addition, β_2 adrenergic signaling capable of inducing RGM-A expression is shown to exert a similar anti-inflammatory effect. The authors conclude that they have identified a mechanism through which the sympathetic nervous system-RGM-a axis modulates resolution of inflammation and tissue repair.

The authors have provided solid evidence supporting a role for RGM-A in immunomodulation. However, the mechanistic links between β_2 signaling, RGM-A and lipid mediators have not been well defined. In light of previous publications by the authors and others that demonstrate RGM-A inhibition of leukocyte migration (PNAS 108:6555) and the involvement of vagus nerve in pro-resolving mediator production (JEM 211:1037) using the same mouse model, additional molecular insights would be required to support the novelty of the current work.

Specific comments:

1. The importance of Fig 1 is unclear (part of Fig. 1e-g have been described in the PNAS paper). Although M-CSF and GM-CSF can push monocyte-derived macrophages to express more M1 or M2 markers, these are not true M1/M2 polarization. More importantly, authors should use data from Fig 5 (which should be validated by Western blot analysis) plus additional mechanistic studies to assess how RGM-A affects macrophage activation and production of pre-resolving lipid mediators.
2. Fig. 2A does not support a role for RGM-A in resolution. It's unclear how R_i is determined. The T50 in Wt cells should be measured at the half of ψ_{max} , which should be about 35 hours, similar to that in Fig. 2f and Fig. 3h. This means that there is no difference in the rate of resolution. To demonstrate effects on resolution and tissue repair, authors need to use relevant models, such as wound healing.
3. To strengthen the link, authors should measure levels RGM-A and lipid mediators in control and RGM-A \pm cells as shown in Fig. 2h and 2i. Similarly, in Fig. 4 RGM-A level needs to be determined in 6-OHD treated mice. The effect of β_2 agonist should also be examined in control and RGM-A \pm mice.
4. As mentioned above, Fig. 5 needs to be validated by western blotting. Specific inhibitors of specific signaling pathways implicated in the mechanism should be applied to determine their involvement in RGM-A signaling.

Reviewer #2 (Autoimmune, neuro-immune crosstalk)(Remarks to the Author):

The manuscript by Korner and colleagues shows the role of RGM-A in regulating resolution of inflammation and tissue repair. While the overall message is clearly delivered, the data presentation is hard to follow.

General comments:

1. Figures are overloaded and confusing.
2. The nomenclature used for defining monocyte-derived macrophages is outdated. The terms M1 and M2 are no longer used. More accurate would be 'M1-like' and 'M2-like', or 'polarized towards pro- or anti-inflammatory activity', respectively.

3. Literature citation is poor in the context of the resolving activities of macrophages and their interactions with the nervous system.

Specific comments:

Figure 1- Too loaded overall, and labeling is not clear.

Figure 1c and Fig 1d- Some groups are repeated; it is not clear which groups were treated with TNF α and which not.

Figure 1f- Not clear which macrophage populations were followed.

Figure 2. What are "classical monocytes"? Does the author mean M1, or undifferentiated monocytes?

Figure 3. The cytokine profile looks the same for the pro-inflammatory cytokine, IL-6, and the resolving cytokine, IL-10. How do the authors explain in such a case the overall resolution?

Figure 3d. What does the author wish to show in this figure?

Figure 5. Not clear how this experiment helps support the overall conclusions.

In particular, we ask that you include new empirical data to provide additional mechanistic insights for the RGM-A/lipid mediator/ β 2AR axis, a critical point raised by referee # 1. We thank the editor and the reviewer for raising this important point. We have performed additional experiments to confirm these data. Since the enzymes 5-LOX and 12/15-LOX contribute to the generation of pro-resolving mediators LXA₄, MaR1 and PDX, we incubated peritoneal macrophages (M Φ) from WT or LOX 12/15^{-/-} mice with RGM-A peptide and found reduced phagocytosis rate of fluorescently labeled ZyA particles after stimulation with RGM-A peptide (**Suppl. Fig. 4a**). In a second set of experiments we incubated human M Φ with RGM-A peptide and the 5- and 12/15-LOX inhibitors baicalein or cinnamyl-3,4-dihydroxy- α -cyamocinnamate (CDC). The impact of RGM-A on M Φ phagocytosis was significantly reduced (**Suppl. Fig. 4b-c**), suggesting the RGM-A effects in resolution to be 5-LOX and 12/15 LOX dependent. To obtain more mechanistic insight in the RGM-A – β ₂AR axis we incubated peritoneal M Φ from RGM-A^{fl/fl}/LysMcre⁻ and RGM-A^{fl/fl}/LysMcre⁺ with the β ₂AR agonist formoterol and determined the phagocytosis rate of fluorescently labeled ZyA particles. Collected results showed that the phagocytosis activity of β ₂AR agonist was significantly reduced in RGM-A^{fl/fl}/LysMcre mice compared to the control group (**Suppl. Fig. 4d**). In a different set of experiments, we incubated human M Φ with β ₂AR agonist and 5- and 12/15-LOX inhibitor baicalein. The gathered data demonstrated a decrease in the phagocytosis rate when compared to the control group (**Suppl. Fig. 4e**). Finally, to validate the synergistic impact described in **Figure 4** we stimulated human M Φ with RGM-A peptide, β ₂AR agonist and baicalein, and found that the phagocytosis rate was significantly reduced (**Suppl. Fig. 4f**). These data confirmed our investigation presented in this manuscript, as both RGM-A and β ₂AR agonist showed pro-resolving impact on acute inflammation. This effect is intensified by the synergistic effect of both substances (**Figure 4**).

We have clarified these points in the revised manuscript page 12, 16, 23 and Suppl. Fig. 4a-f.

1. We also ask that you use additional animal model to better establish the relevance of the proposed RGM-A function in inflammation resolution, another referee #1 point.

We thank the editor and the reviewer for raising this important point. We agree with the editor and the reviewer and have performed additional experiments. In this setting we used a lipopolysaccharide (LPS) induced murine peritonitis model. For this purpose, C57BL/6 mice were injected with LPS and subsequently with RGM-A peptide and in a time series of 4 h, 12 h, 24 h and 48 h we analyzed the dynamic cell composition within the collected peritoneal lavages. Results showed that mice treated with RGM-A peptide demonstrated a strong reduction in leukocyte infiltrates that was combined with a marked decrease in PMN and classical Ly6C^{hi} recruitment when compared with control group (**Suppl. Fig 6a**). Furthermore, RGM-A increased levels of non-classical Ly6C^{lo} monocyte and macrophages that indicated a strong enhancement of the macrophage (MΦ) phagocytosis of apoptotic PMN. Next, we quantified the leukocytes kinetics in the various stages of inflammation and were able to show, based on the resolution indices (R_i)¹, that the R_i interval was reduced from 36 h to 26 h. Next, we investigated the role of RGM-A on the biosynthesis of the lipid resolution-phase mediators using a standardized liquid chromatography tandem mass spectrometry platform (LC-MS/MS). We performed a lipid mediator profile within the collected lavages at two time points - 4 h and 12 h. The obtained data showed that RGM-A induced the biosynthesis of 14,15 EET and MaR1 in the early phase of resolution (4 h post-LPS) and additionally PDX in the later phase (12 h post-LPS). Moreover, we also found well-known precursors for the generation of specialized pro-resolving mediators (SPM) (**Suppl. Fig 6c-d**). Taken together, using an additional well-established LPS induced peritonitis model, we could verify the hypothesis that RGM-A has crucial pro-resolving roles during acute inflammation. We have clarified this point in the revised manuscript. (p. 12-13 and Suppl. Fig. 6)

2. Lastly, please revise data presentation as requested by referee #2. We thank the editor and the reviewer for pointing out these aspects. We have now revised the figures as required by the reviewer.

Reviewer 1

- 1. The importance of Fig 1 is unclear (part of Fig. 1e-g have been described in the PNAS paper). Although M-CSF and GM-CSF can push monocyte-derived macrophages to express more M1 or M2 markers, these are not true M1/M2 polarization. More importantly, authors should use data from Fig 5 (which should be validate by Western blot analysis) plus additional mechanistic studies to assess how RGM-A affects macrophage activation and production of pre-resolving lipid mediators.** We thank the reviewer for pointing out these aspects. Resolution of inflammation is an active process and it is considered to be

a separate process from anti-inflammatory processes^{1,2,3}. An acute inflammatory response is classified into an initiation phase and a resolution phase. The initial phase, which, in its simplest form, is defined by the release of pro-inflammatory mediators that induces particularly the recruitment of PMN. In our previous finding in 2011 (PNAS) we have focused exactly on the effects contributing to the onset of the inflammatory response. Key characteristics of resolution are the cessation of neutrophil infiltration, the counter-regulation of pro-inflammatory mediators, the activation of apoptosis of neutrophil, the enhancement of uptake and clearance of apoptotic cells and microorganisms in inflamed tissues and the biosynthesis of pro-resolving mediators¹. Macrophages have a crucial role in resolution programs, which consequently have a strong promoting influence on wound healing and organ regeneration. In wound healing inflammatory monocytes accumulate in the injured tissue and particularly phagocytosis of tissue debris can induce mononuclear cells to switch from pro-inflammatory to an anti-inflammatory phenotype.

Consequently, as we had mentioned in the manuscript on page 7, it is evident that the monocyte and macrophage lineage is of pivotal importance in tissue homeostasis and the resolution of inflammation^{4,5,6}. Investigations have largely proven that the macrophage phenotype is a result of *differentiation* and *polarization*, depending on the exposed signal^{5,7,8}. In *differentiation*, the interactions between macrophage lineage-differentiation factors, such as GM-CSF or M-CSF, and tissue-specific signals induce the irreversible terminal macrophage state, whereas in *polarization*, mature macrophages respond to particular demands, such as the inflammatory response, by creating reversible polarization states. The phenotype stage can be determined firstly by the analysis of the cell morphology and secondly by the release of the specific markers⁹. That's exactly what we had investigated and demonstrated in **Fig. 1**. Here, we could demonstrate that the activation with RGM-A induced the specific M2 phenotype with a significantly greater number of elongated cell shapes compared to round M1 cells (**Fig. 1b**). Furthermore, when investigating the expression of key genes contributing to M1/M2 *differentiation* (**Fig. 1c**) and *polarization* (**Fig. 1d**), we could show that RGM-A affected both the *differentiation* and the *polarization* of macrophage phenotype by decreasing the levels of M1 markers, such as STAT-1, CD80 and CD40 and significantly increasing the levels of the M2 markers Arg1, CD163 and 206 (**Fig. 1c-d**). In this context, it is to emphasize that these markers are well known as phagocytic receptors and markers of the anti-inflammatory and efferocytic M2 phenotype¹⁰. In the next step we intended to distinguish between M2 and M2-like *polarization* by measuring the release of pro- and anti-inflammatory cytokines. As shown in **Fig. 1d**, RGM-A reduced the expression of IL-1 β and IL-6, whereas IL-10 was significantly increased, suggesting that RGM-A shifted the *polarization* state toward the M2 phenotype (and not M2-like polarization) that is thought to contribute to resolution and metabolic homeostasis⁶.

Concerning the reviewer's comment "***More importantly, authors should use data from Fig 5 (which should be validate by Western blot analysis) plus additional mechanistic studies to assess how RGM-A affects macrophage activation and production of pre-resolving lipid mediators.***" We thank the reviewer for raising this important question. Perhaps the reviewer has missed us demonstrating clearly that in **Fig. 5** we show data collected from a "protein" microarray rather than a gene array. With this array we could noticeably show clearly (on protein level) a broad view on the signaling pathways influenced by which RGM-A and β_2 AR agonist and their mechanistically consequences for the macrophage phenotypes. This microarray delivers valid and significant data, which were analyzed with Ingenuity Pathway Analysis (IPA) - a well-established software. Functional analysis identified most significant canonical pathways and biological functions within the uploaded dataset. Significance of the association between uploaded data and pathways was determined by the ratio of proteins from the dataset divided by the total number of proteins involved in the specific pathways/functions. Additionally, a p-value was calculated using Fisher's exact test. Collected data showed that RGM-A is involved in suppression of NF- κ B activity, which is known to be a crucial transcriptional regulator of the M1 program¹¹ (**Fig. 5a**). Moreover, RGM-A regulates the m-TOR signaling pathway known to be an important driver in regulating macrophage metabolism and functional phenotype by activating RICTOR signaling to promote M2 activation¹¹ (**Fig. 5b**) Furthermore we could show that macrophages treated with RGM-A and the β_2 AR agonist additionally activated the PI3K/AKT pathway, which is important in confining pro-inflammatory responses stepping up anti-inflammatory responses and activating the monocytes/macrophage differentiation and polarization towards a pro-resolving phenotype (**Fig. 5c, Suppl. Fig. 7 and Suppl. Table 5**). With this data, we could show and explain the strong synergistic effects of RGM-A and β_2 AR agonist on pro-resolving processes. We could strengthen these pro-resolving effects when investigating the impact of RGM-A on the expression of the specific G protein-coupled receptors on human macrophages (GPCRs) such as ALX/FPR2 and GPR32 that have been shown to mediate pro-resolving actions¹². We found significantly enhanced human GPR32 and ALX/FPR2 mRNA levels in human macrophages (**Fig. 1i**).

Since the enzymes 5-LOX and 12/15-LOX contribute to the generation of pro-resolving mediators LXA₄, MaR1 and PDX, we incubated peritoneal macrophages (M Φ) from WT or LOX 12/15^{-/-} mice with RGM-A peptide and found reduced phagocytosis rate of fluorescently labeled ZyA particles after stimulation with RGM-A peptide (**Suppl. Fig. 4a**). In a second set of experiments we incubated human M Φ with RGM-A peptide and 5- and 12/15-lipoxygenase inhibitors baicalein or cinnamyl-3,4-dihydroxy- α -cyamocinnamate (CDC). The impact of RGM-A on M Φ phagocytosis was significantly reduced (**Suppl. Fig. 4b-c**) when co-stimulated with 5-LOX and 12/15-LOX inhibitors suggesting the RGM-A effects in resolution to be 5-LOX and 12/15 LOX

dependent. To get more mechanistic insight in the RGM-A – β_2 AR axis we incubated peritoneal M Φ from RGM-A^{fl/fl}/LysMcre⁻ and RGM-A^{fl/fl}/LysMcre⁺ with β_2 AR agonist and determined the phagocytosis rate of fluorescent ZyA particles. Collected results showed that the phagocytosis activity of β_2 AR agonist was significantly reduced in RGM-A^{fl/fl}/LysMcre⁺ mice compared to the control group (**Suppl. Fig. 4d**). Then, we incubated human M Φ with β_2 AR agonist and 5- and 12/15-LOX inhibitor baicalein. The results demonstrated a reduction in the phagocytosis impact compared to the control group (**Suppl. Fig. 4e**). Finally, to validate the synergistic impact described in **Figure 4** we stimulated human M Φ with RGM-A peptide, β_2 AR agonist and baicalein, and found that the phagocytosis rate was significantly decreased (**Suppl. Fig. 4f**). These data confirmed our investigation presented in this manuscript. Both RGM-A and β_2 AR agonist showed pro-resolving impact on acute inflammation. This effect is intensified by the synergistic effect of both substances (**Figure 4**).

We have clarified this point in the revised manuscript pages 12, 16, 23 and Suppl. Fig. 4a-f.

- 2. Fig. 2A does not support a role for RGM-A in resolution. It's unclear how Ri is determined. The T50 in Wt cells should be measured at the half of ψ_{max} , which should be about 35 hours, similar to that in Fig. 2f and Fig. 3h. This means that there is no difference in the rate of resolution. To demonstrate effects on resolution and tissue repair, authors need to use relevant models, such as wound healing.**

We thank the reviewer for this important comment. The resolution phase is defined as the interval from the maximum neutrophil infiltration to the point when it is lost from the tissue (and in parallel, mononuclear cells set in - in a nonphlogistic fashion - and play an important role in tissue repair) we used a well-established ZyA – induced peritonitis model to investigate the impact of RGM-A over time interval (**Scheme 1**)^{13, 14, 15}. The resolution interval has been refined by setting a so-called resolution indices (**Scheme 1**)^{13, 14, 15}. When calculating the resolution indices of two groups and the corresponding comparison, the group with greater inflammation is first used as a reference. Our experiments show that RGM-A^{+/-} mice demonstrated more severe inflammation compared to the control group (**Fig. 2a-c**). Therefore, we focused on the RGM-A^{+/-} mice as reference group and evaluated the corresponding features as follows: We found in RGM-A^{+/-} mice the peak PMN infiltration (Ψ_{max}) to be: $\approx 18 \times 10^6$ and the time point we found this peak (T_{max}) was ≈ 12 h post-ZyA. Then we evaluated 50% of peak PMN (R_{50}) and found R_{50} to be $\approx 9 \times 10^6$ at the time point (T_{50}) of ≈ 37 h post-ZyA. The resolution interval had been calculated as the time between Ψ_{max} and R_{50} , which resulted in a time interval of 25 h (**Fig. 2a**). Consequently, we then calculated from this reference group the parameters in the

control group which showed the following values: $\Psi_{\max} \approx 12 \times 10^6$, $T_{\max} \approx 6$ h, $T_{50} \approx 18$ h, $R_i \approx 12$ h.

Taken together, our findings strongly highlighted that RGM-A either endogenously or exogenously (**Fig. 2**) promote the resolution of inflammation.

Scheme 1 (adapted from¹⁵)

Quantitative definition of exudate resolution and non-resolving inflammation. Hypothetical example of contained self-limited resolving inflammation versus non-resolving inflammation (red line) to illustrate the quantitative indices and components: ψ_{\max} for peak PMN infiltration, 50% of peak PMN (R_{50}), time point of R_{50} (T_{50}), and resolution interval (R_i) to quantify PMN influx and removal as well as non-phlogistic recruitment of monocytes-macrophages in exudates, which is required for repair and renewed function¹⁵.

Regarding the ZyA-induced peritonitis model and the question, whether this offers an adequate model for the description of the resolution mechanisms, I would like to point out that in health, acute inflammatory responses are protective meaning that they are self-limited, in that they resolve on their own to get back to homeostasis¹. This means that acute inflammatory response reflects a temporal dynamic of leukocytes migration, which is divided into the initiation and resolution phase. The intention of this project has been to investigate the influence of RGM-A and adrenergic nerves in resolution processes of acute inflammation. For this purpose,

we used the self-resolving ZyA-induced peritonitis model. Beside various self-resolving models the ZyA-induced peritonitis model is very well established and it has been presented and published in many high impact journals such as ^{16, 17, 18}.

Nevertheless, we agree with the reviewer and have performed additional experiments. In this setting we used a self-resolving lipopolysaccharide (LPS) induced murine peritonitis model. For this purpose, C57BL/6 mice were injected with LPS and subsequently with RGM-A peptide and in a time series of 4 h, 12 h, 24 h and 48 h we analyzed the dynamic cell composition within the collected peritoneal lavages. Results showed that mice treated with RGM-A peptide demonstrated a strong reduction in leukocyte infiltrates that was combined with a marked decrease in PMN and classical Ly6C^{hi} recruitment when compared with control group (**Suppl. Fig. 6a**). Furthermore, RGM-A increased levels of non-classical Ly6C^{lo} monocyte and macrophages that indicated a strong enhancement of the macrophage (MΦ) phagocytosis of apoptotic PMN. Next, we quantified the kinetics of leukocytes in the various stages of inflammation and were able to show, based on the resolution indices (*Ri*), that the *Ri* interval was reduced from 36 h to 26 h. Next, we investigated the role of RGM-A on the biosynthesis of the lipid resolution-phase mediators using a standardized liquid chromatography tandem mass spectrometry (LC-MS/MS). We performed a lipid mediator profile within the collected lavages at two time points - 4 h and 12 h. The obtained data showed that RGM-A induced the biosynthesis of 14,15 EET and MaR1 in the early phase of resolution (4 h post-LPS) and additionally PDX in the later phase (12 h post-LPS). Moreover, we also found well-known precursors for the generation of specialized pro-resolving mediators (SPM) (**Suppl. Fig. 6c-d**). Taken together, using an additional well-established LPS induced peritonitis model, we could verify the hypothesis that RGM-A has crucial pro-resolving roles during inflammation.

We have clarified this point in the revised manuscript. (p. 12-13 and Suppl. Fig. 6)

- 3. To strengthen the link, authors should measure levels RGM-A and lipid mediators in control and RGM-A^{+/-} cells as shown in Fig. 2h and 2i. Similarly, in Fig. 4 RGM-A level needs to be determined in 6-OHD treated mice. The effect of b2 agonist should also be examined in control and RGM-A ^{+/-} mice.** We thank the reviewer for this important comment. As demonstrated in **Suppl. Fig. 1b** we had already studied the RGM-A-mRNA levels in RGM-A^{+/-} where we could show a significant reduction in the gut in RGM-A^{+/-} mice compared to RGM-A^{+/+}. We performed additional experiments to determine the pro-resolving lipid mediators in RGM-A^{+/-} mice. As expected we found a marked increase of TBX₂ and LTB₄ and a strong reduction in the generation of pro-resolving mediators such as PDX and their pathway markers 8-HETE, 12HETE, 7-HDHA, 10-HDHA and 17-HDHA in RGM-A^{+/-} mice compared to RGM-A^{+/+} (**Suppl. Fig. 3c**). To demonstrate the RGM-A expression in 6-OHD mice we performed histological studies, in which we were able to detect a markedly reduced expression

of RGM-A in the neurofilament structures of the peritoneum (**Suppl. Fig. 5**). To strengthen the data for the interaction between RGM-A and β_2 AR agonist, as mentioned above, we incubated peritoneal macrophages from RGM-A^{fl/fl}/LysMcre⁻ and RGM-A^{fl/fl}/LysMcre⁺ with β_2 AR agonist and determined the phagocytosis rate of fluorescently ZYA particles. The gathered results showed that the phagocytosis activity of β_2 AR agonist was significantly reduced in RGM-A^{fl/fl}/LysMcre⁺ mice compared to the control group (**Suppl. Fig. 4d**). Then, we incubated human M Φ with β_2 AR agonist and 5- and 12/15-LOX inhibitor baicalein. The results demonstrated a decrease in the phagocytosis impact compared to the control group (**Suppl. Fig. 4e**). Finally, to validate the synergistic impact described in **Figure 4** we stimulated human M Φ with RGM-A peptide, β_2 AR agonist and baicalein, and found that the phagocytosis rate was strongly reduced (**Suppl. Fig. 4f**). These data confirmed our investigation presented in this manuscript. Both RGM-A and β_2 AR agonist showed a pro-resolving impact on acute inflammation. This effect is intensified by the synergistic effect of both substances (**Figure 4**). We have clarified this point in the revised manuscript. (p.13, 16, 17, Suppl. Fig. 3c, Suppl. Fig. 4d, Suppl. Fig. 5.)

- 4. As mentioned above, Fig. 5 needs to be validated by western blotting. Specific inhibitors of specific signaling pathways implicated in the mechanism should be applied to determine their involvement in RGM-A signaling.** We thank the reviewer for this important comment. As discussed above, perhaps the reviewer has missed us demonstrating clearly that in **Fig. 5** we show data collected from a “protein” microarray and not from a gene array. With this array we could show clearly (on protein level) a broad view on signaling pathways by which RGM-A and β_2 AR agonist mechanistically affect macrophage phenotypes. This microarray delivers valid and significant data, which were analyzed with IPA - a well-established software. Functional analysis identified most significant canonical pathways and biological functions within the uploaded dataset. Significance of the association between uploaded data and pathways was determined by the ratio of proteins from the dataset divided by the total number of proteins involved in the specific pathways/functions. Additionally, a p-value was calculated using Fisher’s exact test. The obtained data showed RGM-A is involved in suppression of NF- κ B activity, which is known to be a crucial transcriptional regulator of the M1 program¹¹ (**Fig. 5a**). Moreover, RGM-A regulates the m-TOR signaling pathway known to be an important driver in regulating macrophage metabolism and functional phenotype by activating RICTOR signaling to promote M2 activation¹¹ (**Fig. 5b**) Furthermore we could show that macrophages treated with RGM-A and the β_2 AR agonist additionally activated the PI3K/AKT pathway, which is important in confining pro-inflammatory responses stepping up anti-inflammatory responses and activating the monocytes/macrophage differentiation and polarization towards a pro-resolving phenotype (**Fig. 5c, Suppl. Fig. 7 and Suppl. Table 5**). With this data, we

could show and explain clearly the strong synergistic effects of RGM-A and β_2 AR agonist on resolution processes.

Reviewer 2

1. Figure 1- Too loaded overall, and labeling is not clear.

We thank the reviewer for raising this point, Figures were rearranged to allow a better traceability of results.

2. Figure 1c and Fig 1d- Some groups are repeated; it is not clear which groups? were treated with TNF α and which not.

We thank the reviewer for raising this point. As we mentioned above and in the manuscript on page 7, it is evident that the monocyte and macrophage lineage is of pivotal importance in tissue homeostasis and the resolution of inflammation^{4, 5, 6}. Investigations have largely proven that the macrophage phenotype is a result of *differentiation* and *polarization*, depending on the exposed signal^{5, 7, 8}. In *differentiation*, the interactions between macrophage lineage-differentiation factors, such as GM-CSF or M-CSF, and tissue-specific signals induce the irreversible terminal macrophage state, whereas in *polarization*, mature macrophages respond to particular demands, such as the inflammatory response, by creating reversible polarization states. In **Figure 1c**, monocytes were differentiated for 7 days either with GM-CSF, M-CSF or RGM-A and PCR analysis was carried out. In contrast, **Fig. 1d** shows polarization of differentiated M1 macrophages that were stimulated either with TNF- α , RGM-A or TNF- α + RGM-A for 24 hours. In summary, RGM-A showed strong impact on both macrophage differentiation and polarization into M2 macrophages. For better understanding, figures were rearranged accordingly.

3. Figure 1f- Not clear which macrophage populations were followed.

We thank the reviewer for raising this important point. The timing and dynamics of leukocyte responses are thought to be crucial for the progression and resolution of inflammation¹. Because neutrophils are the first cells recruited to the site of inflammation in the early phase and monocytes/macrophages predominate during the resolution of inflammation, we sought to investigate the effect of RGM-A on the regulation of neutrophil and macrophage chemotaxis and chemokinesis. In **Figure 1f** M1 macrophages which were differentiated from peripheral blood monocytes were placed into the central loading chamber. To study directly the migration of M1 macrophages, the positive control "monocyte chemotactic protein-1" (MCP-1), which provides a chemoattractive gradient for M1M Φ , was primarily given in one of the peripheral chambers.

Next, in the remaining chambers, RGM-A was added alone and in combination with MCP, and as a negative control RPMI medium was given in one of the chambers. The cells were incubated at 37° for 8 h. As expected we observed a strong increase of M1 macrophage migration towards the positive control MCP-1. However, when focusing on RGM-A chambers we found a strong reduction in M1 MΦ chemotaxis in the direction of the MCP/RGM-A where the cells migrated directly toward the peripheral chamber. Furthermore, we even had negative scores when we focused on the chamber where only RGM-A was alone. This implies that RGM-A has repulsive effects on classical M1 MΦ. These data thus coincide with the above description of the RGM-A influence on the MΦ phenotype.

In **Figure 1g**, however, the M1 macrophages were preincubated with RGM-A at 37° for 10 h and then placed in the central loading chamber. The same substances were added to the peripheral chambers as in Fig 1f. Notably, the treatment with RGM-A indicated a significant increase in macrophage migration toward the RGM-A gradient, whereas the macrophage migration toward the MCP-1 gradient was not affected, suggesting that RGM-A directly shifted the *polarization* state toward the M2 phenotype (**Fig. 1g**).

4. Figure 2. What are "classical monocytes"? Does the author mean M1, or undifferentiated monocytes?

We thank the reviewer for raising this point, and in accordance with recent literature, monocytes can be subdivided into phenotypic different monocyte subsets. "Classical monocytes" and "non-classical monocytes" show distinct cell surface proteins and can also be identified by flow cytometry, where Ly6C is frequently used^{4, 19, 20}. Accordingly, classical monocytes highly express Ly6C (Ly6C^{hi}) and show proinflammatory and phagocytic functions, whereas non-classical monocytes are Ly6C^{lo} and exhibit a unique ability to actively patrol the vascular endothelium²¹.

5. Figure 3. The cytokine profile looks the same for the pro-inflammatory cytokine, IL-6, and the resolving cytokine, IL-10. How do the authors explain in such a case the overall resolution?

We thank the reviewer for raising this point. Unfortunately, in this case it seems there was a mistake while updating graphs from Prism software. Figures were updated and stimulation with β2 ADR agonist leads to reduction of the proinflammatory cytokine IL-6.

6. Figure 3d. What does the author wish to show in this figure?

We thank the reviewer for this important comment. To investigate the role of sympathetic nervous system in the resolution of acute inflammation, we treated mice exposed to ZyA induced peritonitis with selective β₂AR agonist to mimic activation of adrenergic nerves. Initially, we

performed immunochemistry staining of peritoneum to assess the impact of sympathetic adrenergic nerves on RGM-A expression. We identified increased expression of RGM-A within the neurofilament structures of the peritoneum following the stimulation with β_2 AR agonist, suggesting that there is an interplay between RGM-A and β_2 -adrenergic signaling in murine peritonitis.

7. Figure 5. Not clear how this experiment helps support the overall conclusions.

We thank the reviewer for raising this point. Having shown that RGM-A and β_2 -adrenergic nerves regulate the M Φ phenotype and function *in-vitro* and *in-vivo*, we next sought to gain further insight into mechanisms by which RGM-A and β_2 -adrenergic nerves promote resolution programs. We focused on murine peritoneal M Φ that were collected 12 h after induction of peritonitis and treatment with either RGM-A or RGM-A and β_2 AR agonist for analysis. Analysis revealed RGM-A to suppress NF- κ B activity, which is known to be a crucial transcriptional regulator of the M1 program¹¹ (**Fig. 5a**). Moreover, RGM-A regulates the m-TOR signaling pathway known to be a crucial driver in regulating M Φ metabolism and functional phenotype by activating RICTOR signaling to promote M2 activation¹¹ (**Fig. 5b**) Furthermore, collected data show that M Φ treated with RGM-A and the β_2 AR agonist additionally activated the PI3K/AKT pathway, which is important in restricting pro-inflammatory reactions, intensifying anti-inflammatory responses and activating the monocytes/ M Φ differentiation and polarization towards a pro-resolving phenotype (**Fig. 5c, Suppl. Fig. 7 and Suppl. Table 5**). With this data, we could show and explain clearly the strong synergistic effects of RGM-A and β_2 AR agonist on resolution processes.

We thank the reviewers and editors for their time and very helpful comments improving the presentation of our manuscript and results. We trust that the revised manuscript is now suitable publication in *Nature Communications*, and we look forward to hearing from you at your earliest convenience.

Valbona Mirakaj

References

1. Serhan CN, Levy BD. Resolvins in inflammation: emergence of the pro-resolving superfamily of mediators. *The Journal of clinical investigation* **128**, 2657-2669 (2018).
2. Buckley CD, Gilroy DW, Serhan CN. Proresolving lipid mediators and mechanisms in the resolution of acute inflammation. *Immunity* **40**, 315-327 (2014).
3. Serhan CN, Savill J. Resolution of inflammation: the beginning programs the end. *Nature immunology* **6**, 1191-1197 (2005).
4. Wynn TA, Chawla A, Pollard JW. Macrophage biology in development, homeostasis and disease. *Nature* **496**, 445-455 (2013).
5. Okabe Y, Medzhitov R. Tissue biology perspective on macrophages. *Nature immunology* **17**, 9-17 (2015).
6. Ganeshan K, Chawla A. Metabolic regulation of immune responses. *Annual review of immunology* **32**, 609-634 (2014).
7. Varol C, Mildner A, Jung S. Macrophages: Development and Tissue Specialization. *Annual Review of Immunology Vol 33* **33**, 643-675 (2015).
8. Odegaard JI, Chawla A. Alternative Macrophage Activation and Metabolism. *Annu Rev Pathol-Mech* **6**, 275-297 (2011).
9. McWhorter FY, Wang T, Nguyen P, Chung T, Liu WF. Modulation of macrophage phenotype by cell shape. *Proceedings of the National Academy of Sciences of the United States of America* **110**, 17253-17258 (2013).
10. Pluddemann A, Mukhopadhyay S, Gordon S. Innate immunity to intracellular pathogens: macrophage receptors and responses to microbial entry. *Immunological reviews* **240**, 11-24 (2011).
11. Covarrubias AJ, Aksoylar HI, Horng T. Control of macrophage metabolism and activation by mTOR and Akt signaling. *Seminars in immunology* **27**, 286-296 (2015).
12. Serhan CN. Pro-resolving lipid mediators are leads for resolution physiology. *Nature* **510**, 92-101 (2014).
13. Serhan CN, *et al.* Resolution of inflammation: state of the art, definitions and terms. *FASEB journal : official publication of the Federation of American Societies for Experimental Biology* **21**, 325-332 (2007).
14. Bannenberg GL, *et al.* Molecular circuits of resolution: formation and actions of resolvins and protectins. *J Immunol* **174**, 4345-4355 (2005).
15. Serhan CN, Levy BD. Resolvins in inflammation: emergence of the pro-resolving superfamily of mediators. *The Journal of clinical investigation*, (2018).
16. Li Y, Dalli J, Chiang N, Baron RM, Quintana C, Serhan CN. Plasticity of leukocytic exudates in resolving acute inflammation is regulated by MicroRNA and proresolving mediators. *Immunity* **39**, 885-898 (2013).
17. Mirakaj V, Dalli J, Granja T, Rosenberger P, Serhan CN. Vagus nerve controls resolution and pro-resolving mediators of inflammation. *The Journal of experimental medicine* **211**, 1037-1048 (2014).
18. Schwab JM, Chiang N, Arita M, Serhan CN. Resolvin E1 and protectin D1 activate inflammation-resolution programmes. *Nature* **447**, 869-874 (2007).
19. McWhorter FY, Davis CT, Liu WF. Physical and mechanical regulation of macrophage phenotype and function. *Cell Mol Life Sci* **72**, 1303-1316 (2015).
20. Geissmann F, Manz MG, Jung S, Sieweke MH, Merad M, Ley K. Development of monocytes, macrophages, and dendritic cells. *Science* **327**, 656-661 (2010).
21. Shi C, Pamer EG. Monocyte recruitment during infection and inflammation. *Nature reviews Immunology* **11**, 762-774 (2011).

C
Suppl. Figure 3: RGM-A induces lipid mediator class-switching. c) Heterozygous RGM-A-deficient (RGM-A^{+/-}) mice and their littermate controls were injected with ZyA, and peritoneal lavages were collected at 4 h. Levels of bioactive lipid mediators and precursors including the arachidonic acid (AA), docosahexanoic acid (DHA) and eicosapentaenoic acid (EPA) pathway were quantified by LC-MS/MS-based profiling. The results represent two independent experiments and are expressed as mean \pm SEM, n=4-10 per group, *P<0.05.

Suppl. Figure 3

Suppl. Figure 4: Effect of RGM-A on efferocytosis is LOX-dependent. MΦ from 12/15-LOX^{-/-} mice **a**) or RGM-A^{fl/fl}/LysMcre⁺ mice **d**) and littermate controls were stimulated either with RGM-A or β₂ ADR agonist and phagocytosis of fluorescently labeled ZyA particles was determined. Human MΦ were stimulated with RGM-A and the LOX-inhibitors cinnamyl-3,4-dihydroxy- α -cyanocinnamate (CDC) **b**) or baicalein **c**). In a separate experiment, human MΦ were stimulated with β₂ ADR agonist alone **e**) or in combination with RGM-A **f**) and LOX-inhibitor baicalein. The results are from one representative experiment **e-f**) or represent two independent experiments and are expressed as mean \pm SEM, n=5-10 per group, *P<0.05, **P<0.01, ***P<0.001, ****P<0.0001, One-way ANOVA.

Suppl. Figure 4

Suppl. Figure 5: Chemical sympathectomy suppresses RGM-A expression. WT animals were injected with ZyA and vehicle or 6-hydroxydopamine (6-OHD) for 4 h. The expression of RGM-A within the neurofilament structures of peritoneum was analyzed by immunofluorescence.

Suppl. Figure 5

a**b****c**

Legend: ■ LPS + vehicle, ■ LPS + RGM-A
 ng per 10⁵ cells of peritoneal lavage

Suppl. Figure 6

d

Suppl. Figure 6: Administration of RGM-A dampens acute inflammation and enhances resolution during LPS-induced peritonitis. C57BL/6 mice were injected i.p. with LPS and subsequently i.v. with vehicle or RGM-A. Peritoneal lavages were collected at 4, 12, 24 and 48 hours. Total leukocytes were enumerated by light microscopy, PMN, classical-, non-classical monocytes, peritoneal macrophages and monocyte-derived macrophage efferocytosis were determined by flow cytometry **a**). Resolution indices were calculated as defined by ². **b**). For lipidomics, peritoneal fluids were collected and analyzed using LC-MS/MS at 4 hours **c**) and 12 hours **d**). Lipid mediators and their precursors derived from arachidonic acid (AA), docosahexaenoic acid (DHA) and eicosapentaenoic acid (EPA). The results represent two independent experiments and are expressed as mean ± SEM, n=10-15 per group, **P<0.05; ***P<0.01; ****P<0.001; ****P<0.0001, student's t-test.

Suppl. Figure 6

REVIEWERS' COMMENTS:

Reviewer #1 (Remarks to the Author):

The authors used 12/15-LOX^{-/-} model to further strengthen the link between RGM-A and lipid mediators. However, the question regarding how RGM-A affects macrophage activation and production of pro-resolving lipid mediators has not been addressed.

Specific comments:

1. This reviewer understands Fig. 5 was protein array. Like any of the profiling assays, results need to be validated. The true question is whether Fig. 5 offers any mechanistical insight. If it does, authors should preform experiments to demonstrate the signaling pathways downstream of RGM-A mediating the effects. If not, what's the point of Fig. 5?
2. M1/M2 macrophages are well defined, so does resolution. The authors argue that they were looking at differentiation, rather than polarization. However, there is genetic evidence to indicate GM-CSF/M-CSF are specifically required for M1/M2 differentiation. The so-called resolution indices were used only by the authors' group (and Serhan). The data did support RGM-A suppresses inflammation, but not resolution. The new LPS injection supported this notion. As mentioned before, in vivo wound healing assay, which is easy to do, is a better model to address resolution and repair.

Reviewer #2 (Remarks to the Author):

The authors addressed most of the concern of this reviewer. The manuscript has been improved. The only issue that remained unclear is the multiple names of the macrophages. classical, M1, M2, non classical.

This issue should be better presented. Are both M1 and M2 are subtypes of classical according to the authors?

Reviewer # 1

1. This reviewer understands Fig. 5 was protein array. Like any of the profiling assays, results need to be validated. The true question is whether Fig. 5 offers any mechanistical insight. If it does, authors should preform experiments to demonstrate the signaling pathways downstream of RGM-A mediating the effects. If not, what's the point of Fig. 5?

We thank the reviewer for this important comment. We intended to identify pathways involved and regulated by RGM-A and additionally β_2 adrenergic stimulation. We focused on pathway analysis and tested both protein expression and phosphorylation of these proteins with highly specific antibodies to different phosphorylation sites. The collected data could noticeably show clearly a broad overview on the signaling pathways influenced specifically by RGM-A and β_2 adrenergic stimulation and their mechanistically consequences for the macrophage phenotypes. The obtained data, which are valid and significant were analyzed with Ingenuity Pathway Analysis (IPA) - a well-established software. Functional analysis identified most significant canonical pathways and biological functions within the uploaded dataset. Significance of the association between uploaded data and pathways was determined by the ratio of proteins from the dataset divided by the total number of proteins involved in the specific pathways/functions. Additionally, a p-value for specific groups was calculated using Fisher's exact test. Collected data showed that RGM-A is involved in suppression of NF- κ B activity, which is known to be a crucial transcriptional regulator of the M1 program¹ (**Fig. 5a**). Moreover, RGM-A regulates the m-TOR signaling pathway known to be an important driver in regulating macrophage metabolism and functional phenotype by activating RICTOR signaling to promote M2 activation¹ (**Fig. 5b**). I hope thus to adequately answered your question. In addition, I would like to mention that there are numerous publications showing protein array data without additional validation e.g.^{2,3}.

2. **M1/M2 macrophages are well defined, so does resolution. The authors argue that they were looking at differentiation, rather than polarization. However, there is genetic evidence to indicate GM-CSF/M-CSF are specifically required for M1/M2 differentiation. The so-called resolution indices were used only by the authors' group (and Serhan). The data did support RGM-A suppresses inflammation, but not resolution. The new LPS injection supported this notion. As mentioned before, in vivo wound healing assay, which is easy to do, is a better model to address resolution and repair.**

Thank you for raising these points. We focused on macrophage differentiation and particularly polarization. In Fig. 1 we demonstrated the impact of RGM-A on the differentiation and polarization of human macrophages. In *differentiation*, the interactions between macrophage lineage-differentiation factors, such as GM-CSF or M-CSF, and tissue-specific signals induce the irreversible terminal macrophage state, whereas in *polarization*, mature MΦ respond to particular demands, such as the inflammatory response, by creating reversible polarization states. As recent studies indicate that different cell shapes mark the *differentiation* to the M1 or M2 phenotypes⁴, we stimulated human peripheral blood mononuclear cells (PBMC) with GM-CSF, M-CSF or RGM-A for 7 d and then analyzed the cell morphology (**Suppl. Fig. 1a**). As expected, treatment with GM-CSF induced the M1 phenotype and a specific round shape, whereas M-CSF-activated M2 MΦ showed an elongated morphology (**Fig. 1b**). Surprisingly, activation with RGM-A induced the M2 phenotype with a significantly greater number of elongated cell shapes compared to round M1 cells (**Fig. 1b**). To corroborate these results, we next profiled the expression of key genes contributing to M1/M2 *differentiation*. RGM-A decreased the levels of M1 markers, such as STAT-1 and CD80, and significantly increased the levels of the M2 markers Arg1 and CD163 (**Fig. 1c**), which are phagocytic receptors and markers of the anti-inflammatory and efferocytic M2 phenotype⁵. To investigate whether RGM-A may play a direct role in the phenotypic *polarization* of MΦ, we challenged M1 (GM-CSF cultured) MΦ with RGM-A and subsequently stimulated them with TNF-α or vehicle for 24 h. We observed a significant reduction in the levels of the M1 markers STAT-1 CD40 and CD80, whereas the M2 markers Arg1, CD163 and CD206 were significantly increased by RGM-A (**Fig. 1d**). Next, we sought to distinguish between M2 and M2-like *polarization* by measuring the release of pro- and anti-inflammatory cytokines. As shown in **Fig. 1d**, RGM-A reduced the expression of IL-1β and IL-6, whereas IL-10 was significantly increased, suggesting that RGM-A shifted the *polarization* state toward the M2 phenotype that is thought to contribute to resolution and metabolic homeostasis⁶. Together, these data indicate that RGM-A induced both differentiation and polarization toward the M2 pro-healing and pro-resolving phenotype. Moreover, RGM-A promoted nonphlogistic cell recruitment, a key process in the resolution of acute inflammation. Having demonstrated that RGM-A regulates the MΦ phenotype and

function (**Fig. 1**), we next tested whether β_2 -adrenergic signaling might play a role in the phenotypic *differentiation* or *polarization* of human M Φ . We observed the β_2 AR agonist to be mainly involved in the phenotypic *polarization* toward the M2 M Φ as demonstrated by significant reduction in the levels of the M1 markers STAT-1, CD40, CD80 and IL-6, whereas the M2 markers Arg1, CD163, CD206 and the cytokines IL-10 and TGF- β were significantly increased (**Fig. 3c**). Our in-vivo experiments could confirm these effects particularly in the monocyte/macrophage polarization, where RGM-A and β_2 AR agonist decreased classical Ly6C^{hi} monocytes and increased non-classical Ly6C^{low} monocytes, that finally indicated a strong enhancement of macrophage clearance of apoptotic PMN (**Fig. 2 and Fig. 3g**).

Resolution of inflammation is an active process and it is considered to be a separate process from anti-inflammatory processes^{7, 8}. An acute inflammatory response is classified into an initiation phase and a resolution phase. The initial phase is characterized by the activation of the well-known inflammatory events (e.g. activation of cytokines/chemokines, neutrophil recruitment) whereas the resolution process is defined to serve as agonists to reduce the neutrophil recruitment from inflamed site, to counterregulate pro-inflammatory mediators, activate the apoptosis of neutrophils, promote the clearance of apoptotic cells, microorganisms and cell debris by macrophages in inflamed site. These resolution events are regulated by the temporal biosynthesis of novel chemical specialized pro-resolving mediators (SPMs) namely lipoxins, resolvins, protectins and maresins^{7, 9}.

Our findings demonstrate that RGM-A signaling promotes resolution by affecting all defined resolution key characteristics. We have illustrated that both the endogenous and the exogenous RGM-A promote resolution of acute inflammation through affecting exactly these points e.g. the cessation of neutrophil infiltration, the counter-regulation of pro-inflammatory mediators, the activation of apoptosis of neutrophil, the enhancement of uptake and clearance of apoptotic cells cell debris or microorganisms in inflamed tissues (particularly through the regulation of monocyte/macrophage *differentiation* and *polarization* programs) and the biosynthesis of pro-resolving mediators (**Fig. 1, Fig. 2, Fig. 4, Fig. 5, Fig. 6, Suppl. Fig. 6 and Suppl. Fig. 7**).

3. Moreover, I would like to mention that in addition to my group and (Serhan) – there are numerous publications that quantify resolution processes using resolution indices e.g.^{10, 11, 12}. Regarding the in-vivo models and the question, whether these offers an adequate model for the description of the resolution mechanisms, I would like to point out that in health, acute inflammatory responses are protective meaning that they are self-limited, in that they resolve on their own to get back to homeostasis¹³. This means that acute inflammatory response reflects a temporal dynamic of leukocytes migration, which is divided into the initiation and resolution phase. The intention of this project has been to investigate the influence of RGM-A and adrenergic nerves in resolution processes of acute inflammation. Of course, wound

healing assays are very good models to address resolution and repair, but we intended to use models, which are close to the life threatening illness “sepsis”. For this purpose, firstly we used a well-established self-resolving ZyA induced peritonitis model, which has been presented and published in many high impact journals such as ^{14, 15, 16}. Secondly, in a separate set of experiments we could confirm the impact of RGM-A in resolution processes by using a LPS induced peritonitis model.

Reviewer # 2

The authors addressed most of the concern of this reviewer. The manuscript has been improved. The only issue that remained unclear is the multiple names of the macrophages. classical, M1, M2, non classical. This issue should be better presented. Are both M1 and M2 are subtypes of classical according to the authors?

We thank the reviewer for raising this point, and in accordance with recent literature, monocytes and macrophages can be subdivided into phenotypic different monocyte and macrophage subsets. The various nomenclatures describe specific monocyte/macrophage subtypes that fulfill specific tasks in inflammation and particularly in resolution processes. “Classical monocytes” and “non-classical monocytes” show distinct cell surface proteins and can also be identified by flow cytometry, where Ly6C is frequently used^{17, 18, 19}. Accordingly, classical monocytes highly express Ly6C (Ly6C^{hi}) and show pro-inflammatory and phagocytic functions, whereas non-classical monocytes are Ly6C^{lo} and exhibit a unique ability to actively patrol the vascular endothelium²⁰. As described above, we distinguished into M1 and M2 macrophages by profiling the expression of key genes such as STAT-1, CD40, CD80, CD40, Arg1, CD163 and CD206 contributing to M1 or M2 *differentiation and polarization*. **(Fig. 1)** ^{5, 21}.

We thank the editors and reviewers for their time and very helpful comments improving the presentation of our manuscript and results. We trust that the revised manuscript is now suitable publication in *Nature Communications*, and we look forward to hearing from you at your earliest convenience.

Valbona Mirakaj

References

1. Covarrubias AJ, Aksoylar HI, Horng T. Control of macrophage metabolism and activation by mTOR and Akt signaling. *Seminars in immunology* **27**, 286-296 (2015).
2. Vettorazzi S, *et al.* Glucocorticoids limit acute lung inflammation in concert with inflammatory stimuli by induction of SphK1. *Nature communications* **6**, 7796 (2015).
3. Schlegel M, *et al.* Inhibition of neogenin fosters resolution of inflammation and tissue regeneration. *The Journal of clinical investigation* **128**, 4711-4726 (2018).
4. McWhorter FY, Wang T, Nguyen P, Chung T, Liu WF. Modulation of macrophage phenotype by cell shape. *Proceedings of the National Academy of Sciences of the United States of America* **110**, 17253-17258 (2013).
5. Pluddemann A, Mukhopadhyay S, Gordon S. Innate immunity to intracellular pathogens: macrophage receptors and responses to microbial entry. *Immunological reviews* **240**, 11-24 (2011).
6. Ganeshan K, Chawla A. Metabolic regulation of immune responses. *Annual review of immunology* **32**, 609-634 (2014).
7. Buckley CD, Gilroy DW, Serhan CN. Proresolving lipid mediators and mechanisms in the resolution of acute inflammation. *Immunity* **40**, 315-327 (2014).
8. Serhan CN, Savill J. Resolution of inflammation: the beginning programs the end. *Nature immunology* **6**, 1191-1197 (2005).
9. Basil MC, Levy BD. Specialized pro-resolving mediators: endogenous regulators of infection and inflammation. *Nature reviews Immunology* **16**, 51-67 (2016).
10. Kamaly N, *et al.* Development and in vivo efficacy of targeted polymeric inflammation-resolving nanoparticles. *Proceedings of the National Academy of Sciences of the United States of America* **110**, 6506-6511 (2013).
11. Sun G, *et al.* Hemin impairs resolution of inflammation via microRNA-144-3p-dependent downregulation of ALX/FPR2. *Transfusion*, (2018).
12. Karra L, Haworth O, Priluck R, Levy BD, Levi-Schaffer F. Lipoxin B(4) promotes the resolution of allergic inflammation in the upper and lower airways of mice. *Mucosal immunology* **8**, 852-862 (2015).
13. Serhan CN, Levy BD. Resolvins in inflammation: emergence of the pro-resolving superfamily of mediators. *The Journal of clinical investigation* **128**, 2657-2669 (2018).
14. Li Y, Dalli J, Chiang N, Baron RM, Quintana C, Serhan CN. Plasticity of leukocytic exudates in resolving acute inflammation is regulated by MicroRNA and proresolving mediators. *Immunity* **39**, 885-898 (2013).

15. Mirakaj V, Dalli J, Granja T, Rosenberger P, Serhan CN. Vagus nerve controls resolution and pro-resolving mediators of inflammation. *The Journal of experimental medicine* **211**, 1037-1048 (2014).
16. Schwab JM, Chiang N, Arita M, Serhan CN. Resolvin E1 and protectin D1 activate inflammation-resolution programmes. *Nature* **447**, 869-874 (2007).
17. McWhorter FY, Davis CT, Liu WF. Physical and mechanical regulation of macrophage phenotype and function. *Cell Mol Life Sci* **72**, 1303-1316 (2015).
18. Wynn TA, Chawla A, Pollard JW. Macrophage biology in development, homeostasis and disease. *Nature* **496**, 445-455 (2013).
19. Geissmann F, Manz MG, Jung S, Sieweke MH, Merad M, Ley K. Development of monocytes, macrophages, and dendritic cells. *Science* **327**, 656-661 (2010).
20. Shi C, Pamer EG. Monocyte recruitment during infection and inflammation. *Nature reviews Immunology* **11**, 762-774 (2011).
21. Mantovani A, Biswas SK, Galdiero MR, Sica A, Locati M. Macrophage plasticity and polarization in tissue repair and remodelling. *The Journal of pathology* **229**, 176-185 (2013).